# On Fine-Grained Distinct Element Estimation

**Ilias Diakonikolas** [1]  **Daniel M. Kane** [2]  **Jasper C.H. Lee** [3]  **Thanasis Pittas** [1]  **David P. Woodruff** [4]  **Samson Zhou** [5]

## Abstract

We study the problem of distributed distinct element estimation, where $\alpha$ servers each receive a subset of a universe $[n]$ and aim to compute a $(1 + \varepsilon)$-approximation to the number of distinct elements using minimal communication. While prior work establishes a worst-case bound of $\Theta\left(\alpha \log n + \frac{\alpha}{\varepsilon^2}\right)$ bits, these results rely on assumptions that may not hold in practice. We introduce a new parameterization based on the number $C = \frac{\beta}{\varepsilon^2}$ of pairwise collisions, i.e., instances where the same element appears on multiple servers, and design a protocol that uses only $O\left(\alpha \log n + \frac{\sqrt{\beta}}{\varepsilon^2} \log n\right)$ bits, breaking previous lower bounds when $C$ is small. We further improve our algorithm under assumptions on the number of distinct elements or collisions and provide matching lower bounds in all regimes, establishing $C$ as a tight complexity measure for the problem. Finally, we consider streaming algorithms for distinct element estimation parameterized by the number of items with frequency larger than 1. Overall, our results offer insight into why statistical problems with known hardness results can be efficiently solved in practice.

## 1. Introduction

Estimating the number of distinct elements in a large dataset is a fundamental question that was first introduced by Flajolet and Martin (Flajolet & Martin, 1985) and has subsequently received significant attention, e.g., (Cohen, 1997; Alon et al., 1999; Bar-Yossef et al., 2002; Durand & Flajolet, 2003; Raskhodnikova et al., 2009; Kane et al., 2010; Cormode et al., 2011; Woodruff & Zhang, 2012; 2014; Braverman et al., 2018; Blasiok, 2020; Woodruff & Zhou, 2021; Ajtai et al., 2022; Blocki et al., 2023; Jain et al., 2023; Gribelyuk et al., 2024) due to both the simplicity of the question as well as its wide range of applications. We study the problem of distinct element estimation in a distributed setting, so that there are $\alpha$ servers that each receive a subset of the universe $[n] := \{1, \ldots, n\}$. The goal is for the servers to execute a protocol that can approximate the total number of distinct elements, which is the number of coordinates $j \in [n]$ that appears in at least some server. The protocol should use as small of an amount of total communication as possible, where the total communication is the sum of the sizes of all messages exchanged in the protocol in the worst-case. To capture approximation, for a prescribed accuracy parameter $\varepsilon \geq 0$, the goal is to output a $(1 + \varepsilon)$-approximation to the number of distinct elements. The problem of distinct element estimation across a distributed dataset has a large number of applications, including database design (Finkelstein et al., 1988), data warehousing (Acharya et al., 1999; Gibbons, 2001), network traffic monitoring (Akella et al., 2003; Estan et al., 2003; Liu et al., 2020), internet mapping (Palmer et al., 2001), and online analytic processing (OLAP) (Shukla et al., 1996; Padmanabhan et al., 2003).

In the context of machine learning, distributed distinct element estimation plays a crucial role in many applications where data is distributed across multiple nodes or servers. For instance, in collaborative filtering (Resnick et al., 1994), such as recommendation systems (Aggarwal, 2016; Koren et al., 2009), estimating the distinct preferences or behaviors of users across various platforms requires efficient distributed algorithms. Similarly, in anomaly detection (Chandola et al., 2009), identifying rare or novel events across different data sources—such as network traffic or sensor data—requires tracking unique occurrences without centralized data aggregation. Distributed distinct element estimation is also relevant in federated learning (McMahan et al., 2017), where machine learning models are trained across decentralized devices while keeping data local. Estimating the number of distinct features or labels across distributed devices is essential for improving training efficiency. In large-scale graph analysis (Malewicz et al., 2010; Gonzalez et al., 2014), where nodes or edges are distributed

---

[1]University of Wisconsin-Madison [2]University of California, San Diego [3]University of California, Davis [4]Carnegie Mellon University [5]Texas A&M University. Correspondence to: Ilias Diakonikolas <ilias@cs.wisc.edu>, Daniel M. Kane <dakane@cs.ucsd.edu>, Jasper C.H. Lee <jasper-lee@ucdavis.edu>, Thanasis Pittas <pittas@wisc.edu>, David P. Woodruff <dwoodruf@andrew.cmu.edu>, Samson Zhou <samsonzhou@gmail.com>.

*Proceedings of the $42^{nd}$ International Conference on Machine Learning*, Vancouver, Canada. PMLR 267, 2025. Copyright 2025 by the author(s).

across servers, this problem helps in tasks like counting distinct subgraphs or community structures. Additionally, in streaming data applications (Manku & Motwani, 2002), such as real-time monitoring or natural language processing, estimating the diversity of items in large data streams is essential for efficient data summarization and decision-making.

(Kane et al., 2010; Blasiok, 2020) gave a one-pass streaming algorithm for achieving a $(1 + \varepsilon)$-approximation to the number of distinct elements on a dataset from a universe of size $[n]$, using $O\left(\frac{1}{\varepsilon^2} + \log n\right)$ bits of space. This can be transformed into a distributed protocol across $\alpha$ servers that uses $O\left(\frac{\alpha}{\varepsilon^2} + \alpha \log n\right)$ bits of communication, since each server can locally simulate the streaming algorithm on their dataset and then pass the state of the algorithm to the next server. On the lower bound side, (Cormode et al., 2011) showed that distributed distinct element estimation requires $\Omega(\alpha)$ communication, while (Arackaparambil et al., 2009; Chakrabarti & Regev, 2012) showed a lower bound of $\Omega\left(\frac{1}{\varepsilon^2}\right)$. These lower bounds were then subsequently strengthened by (Woodruff & Zhang, 2012) and finally (Woodruff & Zhang, 2014) for all parameter regimes to $\Omega\left(\frac{\alpha}{\varepsilon^2} + \alpha \log n\right)$, seemingly resolving the problem by showing that the protocol of (Kane et al., 2010; Blasiok, 2020) is optimal.

However, the lower bound instance of (Woodruff & Zhang, 2014) requires a constant fraction of coordinates to appear across a constant fraction of servers, which may be unrealistic in many applications. For example, in traffic network monitoring, suppose each server oversees a flow of communication, corresponding to messages from individuals, so that the coordinates of the universe would correspond to IP addresses of the senders of the messages. Then the lower bound instance of (Woodruff & Zhang, 2014) would require that a constant fraction of IP addresses send messages to a constant fraction of the servers, i.e., it requires a constant fraction of all senders to be high volume. In reality, previous studies have shown that internet traffic patterns (Adamic & Huberman, 2002) often follow a Zipfian distribution, i.e., a polynomial decay law, c.f., Definition A.1.

More generally, it has long been observed that many large datasets across other domains follow a Zipfian distribution. For example, the distribution of words in a natural language (Zipf, 2013), e.g., user passwords (Wang & Wang, 2016; Wang et al., 2017; Blocki et al., 2018; Hou & Wang, 2023), the distribution of degrees in the internet graph (Kleinberg et al., 1999), and the distribution of population sizes (Gabaix, 1999; Rhodes, 2023) have all been commonly observed to follow a Zipfian distribution. Indeed, (Mitzenmacher, 2003) claims that "power law distributions are now pervasive in computer science". Thus it seems natural to ask

*Does the distributed distinct element estimation problem still require $\Omega\left(\frac{\alpha}{\varepsilon^2} + \alpha \log n\right)$ communication across more "realistic" distributions?*

## 1.1. Our Contributions

In this paper, we give a resounding negative answer to the above question, translating to positive algorithmic results that break previous impossibility barriers. We introduce a novel parameterization of the distributed distinct element estimation problem, showing that although previous upper and lower bounds show optimality for the worst-case input, these hardness of approximation results do not necessarily apply across various regimes of our parameterization. Namely, we show that the complexity of the problem can be characterized by the number of *pairwise collisions* in the dataset. Formally, for vectors $v^{(1)}, \dots, v^{(\alpha)} \in \{0,1\}^n$, we define the number of pairwise collisions to be the number of ordered triplets $(a, b, i)$ such that $1 \le a < b \le \alpha$, $i \in [n]$, and $v_i^{(a)} = v_i^{(b)} = 1$. We remark that the assumption that the vectors $v^{(i)}$ are binary is without loss of generality, as it turns out the resulting protocols and reductions will behave the same regardless of whether a server has a single instance or multiple instances of a coordinate. Nevertheless for the sake of completeness, for vectors $v^{(1)}, \dots, v^{(\alpha)} \in \{0, 1, \dots, m\}^n$, we define the number of pairwise collisions to be the number of ordered triplets $(a, b, i)$ such that $1 \le a < b \le \alpha$, $i \in [n]$, and $v_i^{(a)} \ge 1$ and $v_i^{(b)} \ge 1$.

We first show a general protocol for the distributed distinct element estimation problem across general ranges of $F_0(S)$, the number of distinct elements in the dataset $S$ that is the union of all items given to all servers.

**Theorem 1.1.** *Given a dataset $S$ on a universe of size $n$ with $C = \beta \cdot O\left(\min\left(F_0(S), \frac{1}{\varepsilon^2}\right)\right)$ pairwise collisions for a parameter $\beta \ge 1$, distributed across $\alpha$ players, there exists a protocol that computes a $(1 + \varepsilon)$-approximation to $F_0(S)$ with probability at least $\frac{2}{3}$ that uses*

$$O\left(\alpha \log n\right) + O\left(\min\left(F_0(S), \frac{1}{\varepsilon^2}\right)\right) \cdot \sqrt{\beta} \log n$$

*bits of communication.*

Theorem 1.1 shows that the $\Omega\left(\frac{\alpha}{\varepsilon^2}\right)$ lower bound of (Woodruff & Zhang, 2014) need not apply when the number of pairwise collisions is in the range of $o(\alpha^2 \cdot F_0(S))$. That is, the lower bound of (Woodruff & Zhang, 2014) only applies when there is a constant fraction of coordinates that appear across a constant fraction of servers.

In the case where the number of pairwise collisions is less than the number of distinct elements, e.g., $C < F_0(S)$, we can further improve the guarantees of our protocol as follows:

**Theorem 1.2.** *Given a dataset $S$ on a universe of size $n$ with the promise that there are at most $C \leq F_0(S)$ pairwise collisions, distributed across $\alpha$ players, there exists a protocol that uses total communication*

$$\tilde{O}\left(\alpha \log n + \max\left(\frac{1}{F_0(S)}, \varepsilon^2\right) \cdot \frac{C}{\varepsilon^2} \log n\right)$$

*bits, and with probability at least $\frac{2}{3}$, outputs a $(1 + \varepsilon)$-approximation to $F_0(S)$.*

**Theorem 1.2.** *Given a dataset $S$ on a universe of size $n$ with the promise that there are at most $C \leq F_0(S)$ pairwise collisions, distributed across $\alpha$ players, there exists a protocol that uses total communication*

$$\tilde{O}\left(\alpha \log n + \max\left(\frac{1}{F_0(S)}, \varepsilon^2\right) \cdot \frac{C}{\varepsilon^2} \log n\right)$$

*bits, and with probability at least $\frac{2}{3}$, outputs a $(1 + \varepsilon)$-approximation to $F_0(S)$.*

*Proof.* Consider Algorithm 2. Recall that with probability at least $0.99$, $(1 - O(\varepsilon))F_0(S) \leq F_0(S_i) \cdot 2^i \leq (1 + O(\varepsilon))F_0(S)$. Thus it suffices to achieve a $(1 + O(\varepsilon))$ approximation to $F_0(S_i)$. For each $j \in [n]$, let $f_j$ be the number of times $j$ appears in $S_i$. Then we have

$$F_0(S_i) = F_1(S_i) - \min(0, f_1 - 1) - \ldots - \min(0, f_j - 1).$$

Let $t_j = \min(0, f_j - 1)$ for all $j \in [n]$ be the excess mass of $j$, so that

$$F_0(S_i) = F_1(S_i) - (t_1 + \ldots + t_n).$$

Let $\mathcal{E}$ be the event that $X$ is a 4-approximation to $F_0(S_i)$. Since $Z = F_1(S_i)$ in the context of Algorithm 2 and $X$ is a 4-approximation to $F_0(S_i)$ conditioned on $\mathcal{E}$, then it suffices to achieve an additive $\eta \cdot X = O(\varepsilon) \cdot F_0(S_i)$ approximation to $(t_1 + \ldots + t_n)$ for $\eta = \frac{\varepsilon}{10}$.

Observe that the expected value of $W \cdot \frac{1}{p}$ satisfies

$$\mathbb{E}\left[W \cdot \frac{1}{p}\right] = \frac{1}{p} \cdot \sum_{j \in [n]} p \cdot t_j = t_1 + \ldots + t_n.$$

Moreover, we can upper bound the variance

$$\mathbb{V}\left[W \cdot \frac{1}{p}\right] \leq \frac{1}{p^2} \cdot \sum_{j \in [n]} p \cdot (t_j)^2.$$

Since $p = \min\left(1, \frac{100C}{\eta^2 X^2}\right)$ and $(t_1^2 + \ldots + t_n^2) \leq C$, then

$$\mathbb{V}\left[W \cdot \frac{1}{p}\right] \leq \frac{\eta^2 X^2}{100C}\left(t_1^2 + \ldots + t_n^2\right)^2 \leq \frac{\eta^2 X^2}{100}.$$

Hence by Chebyshev's inequality, we have that with probability at least $0.99$, $W \cdot \frac{1}{p}$ provides an additive $\eta \cdot X$ error to $(t_1 + \ldots + t_n)$, conditioned on $\mathcal{E}$. By Lemma 2.4, we have that $\mathbf{Pr}[\mathcal{E}] \geq 0.99$. Thus by a union bound, with probability at least $0.98$, Algorithm 2 outputs a $(1 + \varepsilon)$-approximation to $F_0$. Observe that conditioned on the event $\mathcal{E}$, we have $X \leq O\left(\frac{1}{\varepsilon^2}\right)$. Since the number of pairwise collisions is at most $C$, then $F_1(S_i) \leq X + C$. Let $Y$ denote the number of items from $T$ sent across all players. Then we have $\mathbb{E}[Y] \leq p(X + C)$. We have $p = \min\left(1, \frac{100C}{\eta^2 X^2}\right)$ for $\eta = \frac{\varepsilon}{10}$. Note that then for $F_0(S) = \Omega\left(\frac{1}{\varepsilon^2}\right)$, we have

$$\mathbb{E}[Y] \leq O\left(\frac{C}{\varepsilon^2 X} + \frac{C^2}{\varepsilon^2 X^2}\right).$$

Since $C \leq F_0(S) = O(X)$, then $\mathbb{E}[Y] = O\left(\frac{C}{\varepsilon^2 \cdot F_0(S)}\right)$. Otherwise for $F_0(S) = O\left(\frac{1}{\varepsilon^2}\right)$, we have $\mathbb{E}[Y] = O(C)$. The desired claim then follows from Markov's inequality. $\square$

While ascertaining the number of pairwise collisions itself may be a difficult challenge and possibly lead to a chicken-and-egg problem, we remark that computing a loose upper bound $C$ on the number of collisions can be performed much easier, particularly given distributional or other a priori side information about the number of collisions. For example, additional knowledge about the number of collisions can be collected using previous datasets from a similar source, in a similar vein to the auxiliary input that is often utilized by learning-augmented algorithms (Mitzenmacher, 2018; Balcan, 2020; Mitzenmacher & Vassilvitskii, 2020). We also remark that the number of pairwise collisions is an important statistic in other problems, such as uniformity testing (Diakonikolas et al., 2018; Fischer et al., 2018; Acharya et al., 2019; Meir et al., 2019) and closeness testing (Diakonikolas et al., 2019).

We remark that for the case when the number of servers for each item follows a Zipfian distribution across the $\alpha$ servers, then the total number of pairwise collisions is $C = O(\alpha) \cdot F_0(S)$, provided that the Zipfian exponent is a constant larger than $1$. On the other hand, the number of distinct elements can be substantially larger than $\frac{1}{\varepsilon^2}$, where $\varepsilon$ is the desired accuracy for the output estimate for the number of distinct elements. Our results indicate that in this regime, only $\tilde{O}(\alpha \log n)$ bits of communication suffice, which bypasses the known $\Omega\left(\frac{\alpha}{\varepsilon^2} + \alpha \log n\right)$ lower bounds. In particular, if the number of distinct elements is $O(n)$ and $\varepsilon$ is around $\frac{1}{\sqrt{n}}$, then the lower bounds indicate $\Omega(n)$ communication is necessary, which is substantially worse than our protocol that achieves $\tilde{O}(\alpha \log n)$ communication.

We complement Theorem 1.1 and Theorem 1.2 with a pair of lower bounds matching in $\beta$ and $\frac{1}{\varepsilon}$:

**Theorem 1.3.** *Let* $\beta \in [1, \alpha^2]$. *Given a dataset $S$ with $C = \Omega(\beta \cdot F_0(S))$ pairwise collisions, distributed across $\alpha$ players, any protocol that computes a $(1 + \varepsilon)$-approximation to $F_0(S)$ with probability at least $\frac{2}{3}$ uses $\sqrt{\beta} \cdot \Omega\left(\min\left(F_0(S), \frac{1}{\varepsilon^2}\right)\right)$ communication.*

**Theorem 1.4.** *Given a dataset $S$ with the promise that there are at most $C \in [\varepsilon \cdot F_0(S), F_0(S)]$ pairwise collisions distributed across $\alpha$ players, any protocol that computes a $(1 + \varepsilon)$-approximation to $F_0(S)$ with probability at least $\frac{2}{3}$ uses $\Omega\left(\frac{C}{\varepsilon^2 \cdot F_0(S)}\right)$ communication.*

We remark that Theorem 1.3 follows immediately as a parameterization of a lower bound from (Woodruff & Zhang, 2014), while Theorem 1.4 is perhaps our most technically involved contribution. We recall that well-known results, e.g., (Cormode et al., 2011) additionally show that regardless of the number of pairwise collisions and regardless of $F_0(S)$, any protocol that estimates $F_0(S)$ to a constant factor requires $\Omega(\alpha)$ communication. Thus, Theorem 1.3 and Theorem 1.4 together imply the lower bound results in Table 1.

Moreover, we remark that for the regime where $C < \varepsilon \cdot F_0(S)$, then $F_1(S) = \sum_{i \in [\alpha]} \|v^{(i)}\|_1$ becomes an additive $\varepsilon \cdot F_0(S)$ approximation to $F_0(S)$, and so the players can use $O(\alpha \log n)$ bits of communication to deterministically compute a $(1 + \varepsilon)$-multiplicative approximation to $F_0(S)$.

Our results can be viewed as a first step toward analyzing standard statistical problems with known lower bounds, e.g., (Woodruff & Zhang, 2012; 2014) through the lens of parameterized complexity. Thus our work makes important progress toward a better understanding of natural parameters that explain why these problems are not challenging in practice. We summarize our results for the distributed distinct elements estimation problem in Table 1.

In proving Theorem 1.4, we first show the hardness of approximation for the closely related distributed duplication detection problem, in which the goal is for the $\alpha$ servers to approximate the total number of duplicates, where a duplicate is defined to be a coordinate $j \in [n]$ that appears on at least two distinct servers. For discussion on the applications of the distributed duplication detection problem, see Appendix C.

**Theorem 1.5.** *Let $C$ be an input parameter for the number of duplicates and $\varepsilon \in (0, 1)$ be an accuracy parameter. Suppose there are $\alpha$ players, each receiving a set of at most $s$ items from a universe of size $N = \Omega(s)$. Then any protocol $\Pi$ that with probability at least $\frac{2}{3}$, identifies whether there are fewer than $(1 - \varepsilon) \cdot C$ duplicates or more than $(1 + \varepsilon) \cdot C$ duplicates requires $\Omega(\alpha s)$ communication for $C < \frac{4}{\varepsilon^2}$ and $\Omega\left(\frac{\alpha s}{C\varepsilon^2}\right)$ communication for $C \geq \frac{4}{\varepsilon^2}$.*

We remark that for $\varepsilon = 0$, the lower bound of $\Omega(\alpha s)$ follows

via a simple reduction from previous work on non-promise set disjointness in the coordinator model (Braverman et al., 2013). Thus, the main contribution in Theorem 1.5 is to show that even the problem of approximating the number of duplicates requires a substantial amount of total communication. We also give a simple protocol that uses $O\left(\frac{\alpha s \log \alpha}{C\varepsilon^2}\right)$ bits of communication, showing that Theorem 1.5 is near-optimal.

Further, we remark that given Theorem 1.2, a natural question would be to ask whether the promise of the upper bound on $C$ must be known in advance in order to achieve improved communication bounds, perhaps through a preliminary subroutine to estimate $C$. However, Theorem 1.5 shows that in general, one cannot estimate $C$ using "small" total communication when $C$ is small.

Finally, we complement our theoretical results with a number of empirical evaluations in Section 3. We show that the standard CAIDA dataset, often used to analyze statistics on virtual traffic networks, is surprisingly skewed, allowing our algorithm to outperform the previous worst-case theoretical bounds by several orders of magnitude. While this may be an extreme case, it demonstrates that our algorithm can achieve significantly better performance in practice, aligning with our theoretical guarantees and serving as a proof-of-concept that illustrates the accuracy-vs-communication tradeoffs in real-world scenarios.

**Paper organization.** The remainder of this paper is structured as follows. In Section 2.1, we present a parameterized lower bounds for the distributed distinct element estimation problem, assuming the communication complexity of the so-called GapSet problem. We then show a corresponding upper bound in Section 2.2. We defer the proof of the GapSet problem to Appendix B and Appendix C. We provide our experimental results in Section 3. Finally, we show in Appendix D that both distinct element estimation and norm estimation can similarly be parameterized in the streaming model. To a discussion of the notation as well as relevant background statements, we refer the reader to Appendix A.

## 2. Distributed Distinct Element Estimation

In this section, we study the problem of $F_0$ approximation. In Section 2.1, we prove Theorem 1.3, showing that the communication complexity for the distributed distinct element estimation problem is a function of the number of pairwise collisions distributed across the players. In Section 2.2, we give an algorithm for the distributed distinct element estimation problem that uses total communication which is function of the number of pairwise collisions, i.e., the algorithm corresponding to Theorem 1.2.

|  | $C = \beta \cdot F_0(S), \beta \geq 1$ | |
|---|---|---|
|  | $F_0(S) < \frac{1}{\varepsilon^2}$ | $F_0(S) \geq \frac{1}{\varepsilon^2}$ |
| Theorem 1.1 | $O\left(\alpha \log n + \sqrt{\beta} \cdot F_0(S) \cdot \log n\right)$ | $O\left(\alpha \log n + \frac{\sqrt{\beta}}{\varepsilon^2} \log n\right)$ |
| Theorem 1.3 | $\Omega(\alpha + \sqrt{\beta} \cdot F_0(S))$ | $\Omega\left(\alpha + \frac{\sqrt{\beta}}{\varepsilon^2}\right)$ |
|  | $C = \beta \cdot F_0(S), \beta < 1, C > \varepsilon \cdot F_0(S)$ | |
|  | $F_0(S) < \frac{1}{\varepsilon^2}$ | $F_0(S) \geq \frac{1}{\varepsilon^2}$ |
| Theorem 1.2 | $O\left(\alpha \log n + \frac{\beta}{\varepsilon^2} \log n\right)$ | $O\left(\alpha \log n + \beta \cdot F_0(S) \cdot \log n\right)$ |
| Theorem 1.4 | $\Omega\left(\alpha + \frac{\beta}{\varepsilon^2}\right)$ | $\Omega(\alpha + \beta \cdot F_0(S))$ |

Table 1: A summary of our results for the distributed distinct elements estimation problem on a universe of size $n$ across $\alpha$ servers, parameterized by the number $C$ of collisions across the $\alpha$ servers, and the accuracy parameter $\varepsilon \in (0,1)$.

**Distinct elements estimation.** We now formally define the distributed distinct element estimation problem in the coordinator model of communication, which was introduced by (Dolev & Feder, 1992). There exist $\alpha$ servers with vectors $v^{(1)}, \ldots, v^{(\alpha)} \in \{0,1\}^n$ on a universe of size $[n]$. The vectors define an underlying frequency vector $v = v^{(1)} + \ldots + v^{(\alpha)}$. We interchangeably refer to the servers as either players or parties, including a specific server that is designated as the coordinator for the protocol. Each of the servers have access to private sources of randomness. There is a private channel between every server and the coordinator, but there are no channels between the other players, so all communication must be performed through the coordinator. We assume without loss of generality that the protocol is sequential and round-based, i.e., in each round the coordinator speaks to some number of players and await their responses before initiating the next round. Therefore, the protocol must be self-delimiting so that all parties must know when each message has been completely sent.

Given an accuracy parameter $\varepsilon$, the goal is to perform a protocol $\Pi$ so that the coordinator outputs a $(1+\varepsilon)$-approximation to $F_0(v)$ after the protocol has completed. The communication cost of the protocol $\Pi$ is the total number of bits sent by all parties in the worst-case output. Thus, we remark that up to constants, we obtain the same results for the message-passing model, where servers are allowed to communicate directly with each other.

For both our algorithms and lower bounds, the assumption that each local vector is binary is without loss of generality, because the resulting protocols and reductions will behave the same regardless of whether a server has a single instance or multiple instances of a coordinate.

### 2.1. Lower Bounds for Distributed Distinct Element Estimation

To show Theorem 1.3, our starting point is the lower bound instance of (Woodruff & Zhang, 2014), which first defines a problem called $\mathsf{SUM} - \mathsf{DISJ}$, in which there are $\alpha$ players $P_1, \ldots, P_\alpha$ with inputs $X_1, \ldots, X_k \in \{0,1\}^{tL}$ and a coordinator $C$ and $Y \in \{0,1\}^{tL}$. The vectors $X_1, \ldots, X_\alpha, Y$ are organized into $t$ blocks $X_i^{(j)}, Y^{(j)}$ for $i \in [\alpha]$ and $j \in [t]$, each with $L$ coordinates. The inputs to each block of $X_1$ and $Y$ are randomly generated instances of two-player set disjointness, i.e., there are $t$ instances of set disjointness, each with universe size $L$, generated as follows. For each $i \in [L]$, one of the following events occurs:

- With probability $\frac{1}{4}$, $i$ is given to $X_1$.

- With probability $\frac{1}{4}$, $i$ is given to $Y$.

- With probability $\frac{1}{2}$, $i$ is not given to either $X_1$ or $Y$.

After this process is performed for each $i \in [L]$, a special coordinate $c \in [L]$ is then chosen uniformly at random and the allocations of $c$ are reset, so that initially, $c$ is not given to either $X_1$ or $Y$. Then, one of the following events occurs:

- With probability $\frac{1}{2}$, $c$ is given to both players.

- With probability $\frac{1}{2}$, $c$ is given to neither player.

The inputs $X_2, \ldots, X_\alpha$ are then similarly generated, but conditioned on the value of $Y$, so that each pair $(X_i, Y)$ forms an input to two-player set disjointness. Let $\mathsf{DISJ}(X_i^{(j)}, Y^{(j)}) = 0$ if the special coordinate $c$ is given to neither player, i.e., the instance is disjoint, and let $\mathsf{DISJ}(X_i^{(j)}, Y^{(j)}) = 1$ otherwise. We then define $\mathsf{SUM} - \mathsf{DISJ}(X_1, \ldots, X_\alpha, Y)$ to be $\sum_{i=1}^{\alpha} \sum_{j=1}^{t} \mathsf{DISJ}(X_i^{(j)}, Y^{(j)})$.

(Woodruff & Zhang, 2014) proved that computing an additive $O\left(\sqrt{\alpha t}\right)$ error to $\mathsf{SUM} - \mathsf{DISJ}(X_1, \ldots, X_\alpha, Y)$ requires $\Omega(\alpha t L)$ communication. They then reduced the problem of $F_0$ approximation from $\mathsf{SUM} - \mathsf{DISJ}$ as follows.

Given an instance $X_1, \ldots, X_\alpha, Y$ of $\mathsf{SUM} - \mathsf{DISJ}$, the coordinator creates the indicator vector $Z$ corresponding to

$[tL] \setminus Y$. Observe that for $t = O\left(\frac{1}{\varepsilon^2 \alpha}\right)$ and $tL = \Theta\left(\frac{1}{\varepsilon^2}\right)$, a $(1+\varepsilon)$-approximation to $F_0(X_1 + \ldots + X_\alpha + Z)$ suffices for the coordinator to compute an additive $O\left(\sqrt{\alpha t}\right)$ error to $\mathsf{SUM} - \mathsf{DISJ}(X_1, \ldots, X_\alpha, Y)$, given $Y$.

Crucially, a constant fraction of the items are given to a constant fraction of the players, with constant probability due to the distribution of set disjointness, where each coordinate is given to each player $X_i$ with probability at least $\frac{1}{4}$. Therefore, $\Omega\left(\frac{1}{\varepsilon^2}\right)$ coordinates in the frequency vector $X_1 + \ldots + X_\alpha + Z$ have frequency $\Omega(\alpha)$, with probability at least $0.99$. Thus we have the following:

**Lemma 2.1.** *(Woodruff & Zhang, 2014) Given a dataset $S$ with $F_0(S) = \Omega\left(\frac{1}{\varepsilon^2}\right)$ and $C = \Omega(\alpha^2 \cdot F_0(s))$ pairwise collisions, distributed across $\alpha$ players, any protocol that computes a $(1+\varepsilon)$-approximation to $F_0(S)$ with probability at least $\frac{2}{3}$ uses $\Omega\left(\frac{\alpha}{\varepsilon^2}\right)$ communication.*

In fact, for $\beta < \alpha$, we can embed the same problem across $\beta$ players to obtain the following:

**Corollary 2.2.** *Let $\beta \in [1, \alpha^2]$. Given a dataset $S$ with $F_0(S) = \Omega\left(\frac{1}{\varepsilon^2}\right)$ and $C = \Omega(\beta \cdot F_0(s))$ pairwise collisions, distributed across $\alpha$ players, any protocol that computes a $(1+\varepsilon)$-approximation to $F_0(S)$ with probability at least $\frac{2}{3}$ uses $\Omega\left(\frac{\sqrt{\beta}}{\varepsilon^2}\right)$ communication.*

Similarly, for $F_0(s) = O\left(\frac{1}{\varepsilon^2}\right)$ with $F_0(s) = \Omega(\alpha)$, it follows that for $t = O\left(\frac{F_0(s)}{\alpha}\right)$ and $tL = \Theta(F_0(s))$, a $(1+\varepsilon)$-approximation to $F_0(X_1 + \ldots + X_\alpha + Z)$ suffices for the coordinator to determine $\mathsf{SUM} - \mathsf{DISJ}(X_1, \ldots, X_\alpha, Y)$ up to additive error $O\left(\sqrt{\alpha t}\right)$, given $Y$. Hence, we have:

**Corollary 2.3.** *Let $\beta \in [1, \alpha^2]$. Given a dataset $S$ with $F_0(s) = O\left(\frac{1}{\varepsilon^2}\right)$ and $C = \Omega(\beta \cdot F_0(s))$ pairwise collisions, distributed across $\alpha$ players, any protocol that computes a $(1+\varepsilon)$ to $F_0(S)$ with probability at least $\frac{2}{3}$ uses $\Omega\left(\sqrt{\beta} \cdot F_0(S)\right)$ communication.*

Putting together Corollary 2.2 and Corollary 2.3, we have:

**Theorem 1.3.** *Let $\beta \in [1, \alpha^2]$. Given a dataset $S$ with $C = \Omega(\beta \cdot F_0(S))$ pairwise collisions, distributed across $\alpha$ players, any protocol that computes a $(1+\varepsilon)$-approximation to $F_0(S)$ with probability at least $\frac{2}{3}$ uses $\sqrt{\beta} \cdot \Omega\left(\min\left(F_0(S), \frac{1}{\varepsilon^2}\right)\right)$ communication.*

We now give the proof of Theorem 1.4, assuming the correctness of Theorem 1.5, which we defer to Section C.

**Theorem 1.4.** *Given a dataset $S$ with the promise that there are at most $C \in [\varepsilon \cdot F_0(S), F_0(S)]$ pairwise collisions distributed across $\alpha$ players, any protocol that computes a $(1+\varepsilon)$-approximation to $F_0(S)$ with probability at least $\frac{2}{3}$ uses $\Omega\left(\frac{C}{\varepsilon^2 \cdot F_0(S)}\right)$ communication.*

*Proof.* Suppose $\alpha = O(1)$. Note that a multiplicative

$(1+\varepsilon)$-approximation to $F_0(S)$ is an additive $\varepsilon \cdot F_0(S)$ approximation to $F_0(S)$. Consider the hard instance of Theorem 1.5 and recall that it places the $C$ pairwise collisions across unique coordinates, so that $F_0(S) = F_1(S) - C$. Thus an additive $\varepsilon \cdot F_0(S)$ approximation to $F_0(S)$ is an additive $\varepsilon \cdot F_0(S)$ approximation to $C$, which is also a multiplicative $\left(1 + \frac{\varepsilon \cdot F_0(S)}{C}\right)$-approximation to $C$. Observe that the $\alpha$ players can use $O\left(\log \frac{1}{\varepsilon}\right) = O(C)$ bits of communication to compute $F_1(S)$ exactly. By Theorem 1.5, a multiplicative $\left(1 + \frac{\varepsilon \cdot F_0(S)}{C}\right)$-approximation to $C$ approximation requires $\Omega\left(\frac{C}{\varepsilon^2 \cdot F_0(S)}\right)$ communication. $\square$

### 2.2. Upper Bounds for Distributed Distinct Element Estimation

In this section, we present upper bounds for the distributed distinct element estimation problem. In particular, we describe our algorithm that guarantees Theorem 1.2, which shows that the communication complexity of the problem is paraemterized by the number of pairwise collisions. We first recall the following guarantees for a constant-factor approximation to $F_0(S)$.

**Theorem 2.4.** *(Kane et al., 2010; Blasiok, 2020) There exists an algorithm that outputs a $4$-approximation to the number of distinct elements and uses $O(\alpha \log n)$ bits of communication.*

Now, we prove Theorem 1.1 through Algorithm 1.

---
**Algorithm 1** $(1+\varepsilon)$-approximation to $F_0$

---
**Input:** Items given to $\alpha$ players from a universe of size $[n]$, accuracy parameter $\varepsilon \in (0, 1)$
**Output:** $(1+\varepsilon)$-approximation to the number of distinct items
1: Let $X$ be a 4-approximation to $F_0$ ▷Lemma 2.4
2: Let $i_0$ be the largest integer such that $\frac{X}{2^{i_0}} > \frac{1000}{\varepsilon^2}$
3: $i \leftarrow \max(0, i_0)$
4: Let $T_i$ be a subset of $[n]$ where each item is subsampled with probability $\frac{1}{2^i}$
5: Each player sends their items in $T_i$
6: Let $Z$ be the number of unique sent items
7: **Return** $Z \cdot 2^i$

---

We now show the parameterized complexity of the distributed distinct elements estimation problem.

**Theorem 1.1.** *Given a dataset $S$ on a universe of size $n$ with $C = \beta \cdot O\left(\min\left(F_0(S), \frac{1}{\varepsilon^2}\right)\right)$ pairwise collisions for a parameter $\beta \geq 1$, distributed across $\alpha$ players, there exists a protocol that computes a $(1+\varepsilon)$-approximation to $F_0(S)$ with probability at least $\frac{2}{3}$ that uses*

$$O(\alpha \log n) + O\left(\min\left(F_0(S), \frac{1}{\varepsilon^2}\right)\right) \cdot \sqrt{\beta} \log n$$

*bits of communication.*

*Proof.* Consider Algorithm 1. Let $F_0(S)$ be the number of distinct items across all players. Note that we have $\mathbb{E}\left[Z \cdot 2^i\right] = F_0(S)$ and $\mathbb{V}\left[Z \cdot 2^i\right] = F_0(S) \cdot 2^i \leq \frac{(\varepsilon \cdot F_0(S))^2}{250}$. Hence conditioned on the correctness of $X$, by Chebyshev's inequality, we have that with probability at least 0.99,

$$(1 - \varepsilon) \cdot F_0(S) \leq Z \cdot 2^i \leq (1 + \varepsilon) \cdot F_0(S).$$

It remains to show that the total communication used by the protocol is $O\left(\alpha \log n\right) + O\left(\min\left(F_0(S), \frac{1}{\varepsilon^2}\right)\right) \cdot \sqrt{\beta} \log n$ bits. Conditioned on the correctness of $X$, we have by the definition of $i$ that $\mathbb{E}\left[Z\right] \leq \frac{800}{\varepsilon^2}$. Hence by Markov's inequality, we have that $Z \leq \frac{10^6}{\varepsilon^2}$ with probability at least 0.99. Let $\mathcal{E}$ be the event that $Z \leq \min\left(\frac{10^6}{\varepsilon^2}, F_0(S)\right)$. Let $N = \min\left(\frac{10^6}{\varepsilon^2}, F_0(S)\right)$ and for $i \in [N]$, let $H_i$ be the number of players with item $i$, so that $0 \leq H_i \leq \alpha$. Then conditioned on $\mathcal{E}$, the number of pairwise collisions is $C = \binom{H_1}{2} + \ldots + \binom{H_N}{2}$. Note that $\binom{H}{2} \geq \frac{H^2}{4} - \frac{1}{4}$, so that $C \geq \frac{H_1^2 + \ldots + H_N^2}{4} - N$. By the Root-Mean Square-Arithmetic Mean Inequality, we have that if there are $C = \beta \cdot O\left(\min\left(F_0(S), \frac{1}{\varepsilon^2}\right)\right)$ pairwise collisions, then

$$H_1 + \ldots + H_N = O\left(\sqrt{\beta}N\right)$$
$$= O\left(\min\left(F_0(S), \frac{1}{\varepsilon^2}\right)\right) \cdot \sqrt{\beta}.$$

Thus the $\alpha$ players have at most $O\left(\min\left(F_0(S), \frac{1}{\varepsilon^2}\right)\right) \cdot \sqrt{\beta}$ items in $S_i$, from a universe of size $[n]$, so the communication for sending these items is $O\left(\alpha \log n\right) + O\left(\min\left(F_0(S), \frac{1}{\varepsilon^2}\right)\right) \cdot \sqrt{\beta} \log n$ bits. Finally, recall from Lemma 2.4 that $O\left(\alpha \log n\right)$ bits of communication suffices to compute a constant-factor approximation to $F_0(S)$. Thus, the total communication is at most $O\left(\alpha \log n\right) + O\left(\min\left(F_0(S), \frac{1}{\varepsilon^2}\right)\right) \cdot \sqrt{\beta} \log n$ bits. $\square$

The guarantees of Theorem 1.2 then follow from Algorithm 2:

**Theorem 1.2.** *Given a dataset $S$ on a universe of size $n$ with the promise that there are at most $C \leq F_0(S)$ pairwise collisions, distributed across $\alpha$ players, there exists a protocol that uses total communication*

$$\tilde{O}\left(\alpha \log n + \max\left(\frac{1}{F_0(S)}, \varepsilon^2\right) \cdot \frac{C}{\varepsilon^2} \log n\right)$$

*bits, and with probability at least $\frac{2}{3}$, outputs a $(1 + \varepsilon)$-approximation to $F_0(S)$.*

---

**Algorithm 2** $(1 + \varepsilon)$-approximation to $F_0$, given an upper bound on the number of collisions

**Input:** Items given to $\alpha$ players from a universe of size $[n]$, accuracy parameter $\varepsilon \in (0, 1)$, upper bound $C$ on the number of pair-wise collisions
**Output:** $(1 + \varepsilon)$-approximation to the number of distinct items
1: Let $X$ be a 4-approximation to $F_0$        ▷Lemma 2.4
2: Let $i_0$ be the largest integer such that $\frac{X}{2^{i_0}} > \frac{1000}{\varepsilon^2}$
3: $i \leftarrow \min(0, i_0)$
4: Let $S_i$ be a subset of $[n]$ where each item is subsampled with probability $\frac{1}{2^i}$
5: Assume without loss of generality each player $i$ has a binary vector $v^{(i)} \in \{0, 1\}^n$
6: Each player sends their total number of items in $S_i$
7: Let $Z$ be the sum of these numbers
8: $\eta \leftarrow \frac{\varepsilon}{10}, p \leftarrow \min\left(1, \frac{100C}{\eta^2 X^2}\right)$
9: Let $T$ be a subset of $S_i$ where each item is subsampled with probability $p$
10: Each player sends their items in $T$
11: Let $W = \sum_{j \in T} \max(0, v_j - 1)$, where $v = \sum_{i \in [\alpha]} v^{(i)}$ be the excess mass in $T$
12: **Return** $Z \cdot 2^i - W \cdot \frac{1}{p}$

---

*Proof.* Consider Algorithm 2. Recall that with probability at least 0.99, $(1 - O(\varepsilon))F_0(S) \leq F_0(S_i) \cdot 2^i \leq (1 + O(\varepsilon))F_0(S)$. Thus it suffices to achieve a $(1 + O(\varepsilon))$ approximation to $F_0(S_i)$. For each $j \in [n]$, let $f_j$ be the number of times $j$ appears in $S_i$. Then we have

$$F_0(S_i) = F_1(S_i) - \min(0, f_1 - 1) - \ldots - \min(0, f_j - 1).$$

Let $t_j = \min(0, f_j - 1)$ for all $j \in [n]$ be the excess mass of $j$, so that

$$F_0(S_i) = F_1(S_i) - (t_1 + \ldots + t_n).$$

Let $\mathcal{E}$ be the event that $X$ is a 4-approximation to $F_0(S_i)$. Since $Z = F_1(S_i)$ in the context of Algorithm 2 and $X$ is a 4-approximation to $F_0(S_i)$ conditioned on $\mathcal{E}$, then it suffices to achieve an additive $\eta \cdot X = O(\varepsilon) \cdot F_0(S_i)$ approximation to $(t_1 + \ldots + t_n)$ for $\eta = \frac{\varepsilon}{10}$.

Observe that the expected value of $W \cdot \frac{1}{p}$ satisfies

$$\mathbb{E}\left[W \cdot \frac{1}{p}\right] = \frac{1}{p} \cdot \sum_{j \in [n]} p \cdot t_j = t_1 + \ldots + t_n.$$

Moreover, we can upper bound the variance

$$\mathbb{V}\left[W \cdot \frac{1}{p}\right] \leq \frac{1}{p^2} \cdot \sum_{j \in [n]} p \cdot (t_j)^2.$$

Since $p = \min\left(1, \frac{100C}{\eta^2 X^2}\right)$ and $(t_1^2 + \ldots + t_n^2) \leq C$, then

$$\mathbb{V}\left[W \cdot \frac{1}{p}\right] \leq \frac{\eta^2 X^2}{100C}\left(t_1^2 + \ldots + t_n^2\right)^2 \leq \frac{\eta^2 X^2}{100}.$$

Hence by Chebyshev's inequality, we have that with probability at least $0.99$, $W \cdot \frac{1}{p}$ provides an additive $\eta \cdot X$ error to $(t_1 + \ldots + t_n)$, conditioned on $\mathcal{E}$. By Lemma 2.4, we have that $\mathbf{Pr}[\mathcal{E}] \geq 0.99$. Thus by a union bound, with probability at least $0.98$, Algorithm 2 outputs a $(1 + \varepsilon)$-approximation to $F_0$. Observe that conditioned on the event $\mathcal{E}$, we have $X \leq O\left(\frac{1}{\varepsilon^2}\right)$. Since the number of pairwise collisions is at most $C$, then $F_1(S_i) \leq X + C$. Let $Y$ denote the number of items from $T$ sent across all players. Then we have $\mathbb{E}[Y] \leq p(X + C)$. We have $p = \min\left(1, \frac{100C}{\eta^2 X^2}\right)$ for $\eta = \frac{\varepsilon}{10}$. Note that then for $F_0(S) = \Omega\left(\frac{1}{\varepsilon^2}\right)$, we have

$$\mathbb{E}[Y] \leq O\left(\frac{C}{\varepsilon^2 X} + \frac{C^2}{\varepsilon^2 X^2}\right).$$

Since $C \leq F_0(S) = O(X)$, then

$$\mathbb{E}[Y] = O\left(\frac{C}{\varepsilon^2 \cdot F_0(S)}\right).$$

Otherwise for $F_0(S) = O\left(\frac{1}{\varepsilon^2}\right)$, we have $\mathbb{E}[Y] = O(C)$. The desired claim then follows from Markov's inequality. $\square$

## 3. Empirical Evaluations

In this section, we describe our empirical evaluations for evaluating our distributed protocol for distinct element estimation. We used the CAIDA dataset (CAIDA, 2016), which consists of anonymized passive traffic traces collected from the high-speed monitor at the "equinix-nyc" data center. This dataset is widely used for statistical analyses for traffic network monitoring, in particular empirical analyses of algorithms for distinct element estimation, norm and frequency moments, and heavy-hitters (Hsu et al., 2019; Chen et al., 2022; Lin et al., 2022; Ivkin et al., 2022). From 12 minutes of internet flow data totaling approximately 40 million total events, we extracted the first 1 million events, each representing an interaction between a sender IP address and a receiver IP address.

**Experimental setup.** We estimate the total number of distinct sender IP addresses. As the events are partitioned across different receiver IP addresses, each receiver holds a different set of users from the total collection of active sender IP addresses. To show our setting is valid for our theoretical assumptions, we considered two different distributions. First, we computed the number of unique senders per receiver and plotted a logarithmic scale of the resulting distribution in Figure 1a. We then isolated the receiver

with the most activity and computed the number of interactions per sender to that IP address, plotting the resulting distribution in Figure 1b.

We then evaluate our distributed protocol in Algorithm 1. In particular, we consider the total communication of our algorithm compared to the total communication given by the analysis of the standard protocol, which sends a sketch of size $O\left(\frac{1}{\varepsilon^2}\right)$ for each of the $\alpha$ servers. Correspondingly, we set our algorithm to also have accuracy $O(\varepsilon)$ and compare the communication, across various values of $\varepsilon = \frac{1}{2^p}$, with $p \in \{2, 3, 4, 5, 6, 7, 8, 9, 10, 11\}$. These results appear in Figure 2a. Finally, we studied the accuracy of our distributed protocol. We evaluated the output of our algorithm for $\varepsilon = \frac{1}{2^p}$, across $p \in \{0, 1, 2, 3, 4, 5\}$ and computed the error with respect to the true number of unique sender IP addresses, which totaled 42200. We give these results in Figure 2b.

Our empirical evaluations were performed with Python 3.11.5 on a 64-bit operating system on an Intel(R) Core(TM) i7-3770 CPU, with 16GB RAM and 4 cores with base clock 3.4GHz. The code is publicly available at `https://github.com/samsonzhou/DKLPWZ25`.

**Results and discussion.** As virtual traffic is generally known to be dominated by a few heavy-hitters, it was not altogether surprising that the distributions of activity for both receiver IP addresses and sender IP addresses were skewed. However, it was a bit surprising that when fit to a Zipfian power law so that the frequency of the $i$-th most common interaction is roughly $\frac{C}{i^s}$, the receiver IP address distribution returned roughly $s \approx 0.743$ and $C \approx 1404.68$, which indicated a highly skewed distribution. By comparison, the activity distribution returned a more modest $s \approx 0.344$ and $C \approx 43.93$. Indeed, we emphasize that although both graphs in Figure 1 appear linear, the scale for the receiver distribution is actually logarithmic.

Because the distribution is so skewed, the number of pairwise collisions is quite small, as most of the receiver IP addresses only receive a small amount of activity. Therefore, our protocol vastly outperforms the theoretical bounds for the standard benchmark by several orders of magnitude, as evident in Figure 2a. Moreover, our algorithm quickly converges to the optimal solution as $\varepsilon$ decreases, achieving 70% error for $\varepsilon = 1$, quickly up to more than 95% error for $\varepsilon = \frac{1}{16}$ in Figure 2b. This matches our theoretical guarantees, thus serving as a simple proof-of-concept demonstrating the accuracy-vs-communication tradeoffs.

## 4. Conclusion

In conclusion, this paper addresses the distributed distinct element estimation problem, where previous results indi-

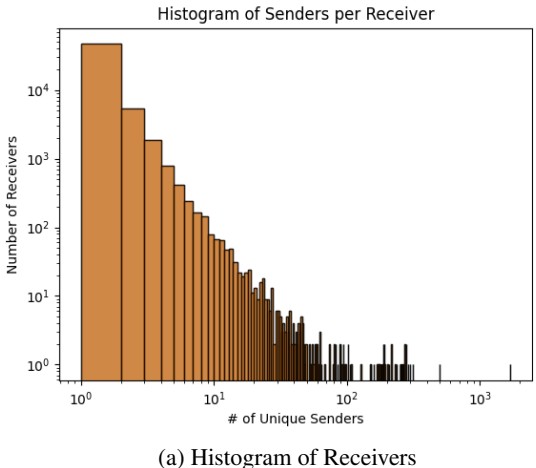

(a) Histogram of Receivers

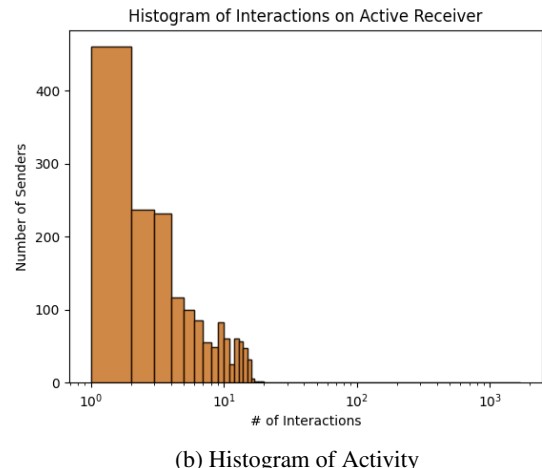

(b) Histogram of Activity

Figure 1: Histogram of unique senders per receiver in Figure 1a. Histogram of activity per sender in most active receiver Figure 1b.

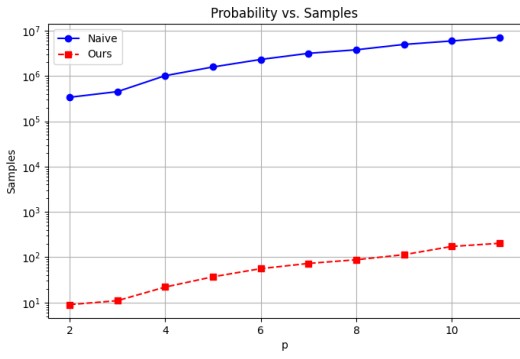
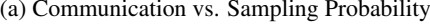

(a) Communication vs. Sampling Probability

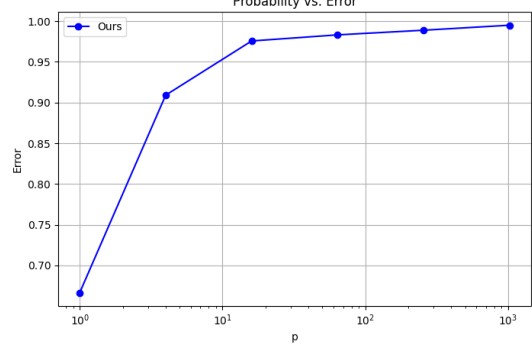

(b) Sampling Probability vs. Accuracy

cate that $\Theta\left(\alpha \log n + \frac{\alpha}{\varepsilon^2}\right)$ bits of communication are both necessary and sufficient in the worst case. However, the assumption of large input sizes across many servers can be unrealistic in practical scenarios. To address this, we introduce a new parameterization based on the number $C$ of pairwise collisions distributed across the $\alpha$ players. Our algorithm, which uses $O\left(\alpha \log n \log \log n + \frac{\sqrt{C}}{\varepsilon} \log n\right)$ bits of communication, demonstrates that small values of $C$ can break existing lower bounds. We also establish matching lower bounds for all regimes of $C$, showing that it provides a tight characterization of the communication complexity for this problem. Ultimately, our work offers new insights into why standard statistical problems, despite known impossibility results, can be efficiently tackled in real-world scenarios.

## Acknowledgments

Ilias Diakonikolas is supported by NSF Medium Award CCF-2107079 and an H.I. Romnes Faculty Fellowship. The work of Jasper C.H. Lee was done in part while he was at UW Madison, supported by NSF Medium Award CCF-2107079. Thanasis Pittas is supported by NSF Medium Award CCF-2107079. Daniel Kane is supported by by NSF Medium Award CCF-2107547 and NSF CAREER Award CCF-1553288. David P. Woodruff is supported in part by Office of Naval Research award number N000142112647 and a Simons Investigator Award. The work was conducted in part while David P. Woodruff and Samson Zhou were visiting the Simons Institute for the Theory of Computing as part of the Sublinear Algorithms program. Samson Zhou is supported in part by NSF CCF-2335411.

## Impact Statement

This paper presents work whose goal is to advance the theoretical foundations of distributed computing. There are many potential societal consequences of our work, none which we feel must be specifically highlighted here.

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

# A. Preliminaries

For a positive integer $n > 0$, we use the notation $[n]$ to represent the set $\{1, 2, \ldots, n\}$. We use $\mathrm{polylog}(n)$ to denote a fixed polynomial in $\log n$.

For a vector $v \in \mathbb{R}^n$, we define $F_0(v) = |\{i \in [n] \mid v_i \neq 0\}|$ and $F_1(v) = |v_1| + \ldots + |v_n|$. For a random variable $X$, we use $\mathbb{E}[X]$ to denote its expectation and $\mathbb{V}[X]$ to denote its variance.

**Definition A.1** (Zipfian distribution dataset)**.** *We say a sequence $X = \{x_1, \ldots, x_n\}$ follows a Zipfian distribution with exponent $s$ if there exist parameters $C_1, C_2 > 0$ such that for any index $i$, we have $\frac{C_1}{i^s} \leq x_i \leq \frac{C_2}{i^s}$.*

Recall the following definition of the squared Hellinger distance.

**Definition A.2** (Squared Hellinger distance)**.** *For two distributions $P$ and $Q$ with probability density functions $f$ and $g$, respectively, defined on a space $X$, their squared Hellinger distance is defined by*

$$h^2(P, Q) = \frac{1}{2} \int_X \left( \sqrt{f(x)} - \sqrt{g(x)} \right)^2 \, dx$$

It can be shown, c.f., [Lemma B.3](#) in [Section B](#), that the squared Hellinger distance between a function on two random variables is a lower bound on informally the mutual information between one of the random variables and the corresponding value of the function on that random variable.

**Communication complexity.** We now recall some preliminaries from communication and information complexity.

**Definition A.3** (Entropy, conditional entropy, mutual information)**.** *Given a pair of random variables $X$ and $Y$ with joint distribution $p(x, y)$ and marginal distributions $p(x)$ and $p(y)$, the* entropy *of $X$ is defined as $H(X) := -\sum_x p(x) \log p(x)$. The* conditional entropy *is $H(X|Y) := -\sum_{x,y} p(x, y) \log \frac{p(y)}{p(x,y)}$. The* mutual information *is $I(X; Y) := H(X) - H(X|Y) = \sum_{x,y} p(x, y) \log \frac{p(x,y)}{p(x)p(y)}$.*

**Definition A.4** (Information cost)**.** *Let $\Pi$ be a randomized protocol that produces a (possibly random) transcript $\Pi(X_1, \ldots, X_T)$ on inputs $X_1, \ldots, X_T$ drawn from a distribution $\mu$. The* information cost *of $\Pi$ with respect to $\mu$ is $I(P_1, \ldots, P_T; \Pi(P_1, \ldots, P_T))$.*

**Fact A.5** (Information cost to communication complexity)**.** *For any distribution $\mu$ and failure probability $\delta \in (0, 1)$, the communication cost of any randomized protocol for $\mu$ on a problem $f$ that fails with probability $\delta$ is at least the information cost of $f$ under distribution $\mu$ and failure probability $\delta$.*

**Definition A.6** (Conditional information cost)**.** *Let $\Pi$ be a protocol on $((\mathbf{x}, \mathbf{y}), R) \sim \eta$ for $\mathbf{x} \sim X$ and $\mathbf{y} \sim Y$, where $\eta$ is a mixture of product distributions on $X^n \times Y^n \times \mathcal{R}$ and $R \sim \mathcal{R}$ is a source of randomness. Then we define the conditional information cost of $\Pi$ with respect to $\eta$ by $I(\mathbf{x}, \mathbf{y}; \Pi(\mathbf{x}, \mathbf{y})|R)$.*

**Definition A.7** (Conditional information complexity)**.** *Given a failure probability $\delta \in (0, 1)$ and a mixture $\eta$ of product distributions, we define the conditional information complexity of $f$ with respect to $\eta$ as the minimum conditional information cost of a protocol for $f$ with failure probability at most $\delta$, with respect to $\eta$, i.e.,*

$$\mathsf{CIC}_{\eta, \delta}(f) = \min_{\Pi} I(\mathbf{x}, \mathbf{y}; \Pi(\mathbf{x}, \mathbf{y})|R),$$

*where the minimum is taken over all protocols $\Pi$ with failure probability at most $\delta$ on the distribution $\eta$.*

**Lemma A.8** (Proposition 4.6 in ([Bar-Yossef et al., 2004](#)))**.** *Let $\mu$ be a distribution on $X^n \times Y^n \times$. If $\eta$ is a mixture of product distributions on $X^n \times Y^n \times \mathcal{R}$ such that the marginal distribution on $X^n \times Y^n$ is $\mu$, then the information cost of a function $f$ with success probability $1 - \delta$ on $\mu$ is at least $\mathsf{CIC}_{\eta, \delta}(f)$.*

**Fact A.9** (Chain rule)**.** *Given discrete random variables $X, Y, Z$, then*

$$I(X, Y; Z) = I(X; Z) + I(X; Y|Z).$$

**Fact A.10** (Maximum likelihood estimation principle)**.** *Let $X \in \mathcal{X}$ and $Y \in \mathcal{Y}$ be randomly selected from some underlying distribution $\mu$. Then there exists a deterministic function $g : Y \to X$ with error $\delta \leq 1 - \frac{1}{2^{H(X|Y)}}$.*

**Definition A.11.** *For a vector $\mathbf{y} \in X^n$, let $j \in [n]$ and $x \in X$. We define $\mathsf{Embed}(\mathbf{y}, j, x)$ to be the $n$-dimensional vector $\mathbf{y}$ with its $j$-th coordinate replaced by $x$, i.e., for $\mathbf{z} = \mathsf{Embed}(\mathbf{y}, j, x)$, we have $z_i = y_i$ for $i \neq j$ and $z_j = x$.*

**Definition A.12** (Decomposable function). *Let $f : X^n \to \{0, 1\}$ be a function. Then we say $f$ is $g$-decomposable with primitive $h$ if there exist functions $g : \{0, 1\}^n \to \{0, 1\}$ and $h : X \to \{0, 1\}$ such that $f(\mathbf{x}, \mathbf{y}) = g(h(x_1, y_1), \ldots, h(x_n, y_n))$.*

**Definition A.13** (Collapsing distribution). *Let $f : X^n \to \{0, 1\}$ be $g$-decomposable with primitive $h$. We say that $(\mathbf{w}, \mathbf{z}) \in X^n$ is a collapsing input for $f$ if for every $j \in [n]$ and $x, y \in X$, we have $f(\mathsf{Embed}(\mathbf{w}, j, x), \mathsf{Embed}(\mathbf{z}, j, y)) = h(x, y)$. We call a distribution $\mu$ on $X^n$ a collapsing distribution for $f$ if every $(\mathbf{w}, \mathbf{z})$ in the support of $\mu$ is a collapsing input.*

**Theorem A.14** (Direct sum, Theorem 5.6 in (Bar-Yossef et al., 2004)). *Let $f : X^n \to \{0, 1\}$ be a decomposable function with primitive $h$ and let $\zeta$ be a mixture of product distributions on $X \times \mathcal{D}$. Let $\eta = \zeta^n$ and $((\mathbf{x}, \mathbf{y}), \mathbf{D}) \sim \eta$. Then if the distribution of $(\mathbf{x}, \mathbf{y})$ is a collapsing distribution for $f$, we have $\mathsf{CIC}_{n,\delta}(f) \geq n \cdot \mathsf{CIC}_{\zeta,\delta}(h)$.*

## A.1. Technical Overview

In this section, we describe the intuition behind our algorithms and lower bounds for the distributed distinct elements estimation problem.

### A.1.1. Protocols for Distributed Distinct Element Estimation

We first describe our general protocol for the distributed distinct element estimation problem across general ranges of $F_0(S)$, the number of distinct elements in the dataset $S$ that is the union of all items given to all servers, i.e., Theorem 1.1.

**Constant-factor approximation.** As a standard subroutine, our algorithm first computes a constant factor approximation to the number of distinct elements. Recall that this is done by subsampling the universe $[n]$ at less and less aggressive rates. The $\alpha$ servers jointly set $S_0 = [n]$ and for each $i \geq 1$, the servers use public randomness to jointly sample each element of $S_{i-1}$ into $S_i$ with probability $\frac{1}{2}$. For example, the expected number of elements in $S_1$ is $\frac{n}{2}$ and so forth. The servers initialize $i = \lceil \log n \rceil$ and send all of their local items that are contained within $S_i$ to a designated server, which is marked as a coordinator. They then send all of their items that are contained in $S_{i-1}$ and so forth, until the coordinator sees $\Theta(1)$ distinct elements across the entire set of items sent from all servers. Using a standard expectation and variance technique, it follows that rescaling the number of distinct elements seen by the coordinator by $\frac{1}{p}$, where $p$ is the sampling probability of the universe induced by $S_i$, is a constant-factor approximation to $F_0(S)$. The total communication used by this protocol is $O(\alpha \log n)$ for the $\alpha$ parties to report the identities of $O(1)$ items across the sets $S_i, S_{i-1}, \ldots$ before the algorithm terminates, combined with an additional $\log \log n$ overhead to handle a naïve union bound over at most $O(\log n)$ possible such sets, i.e., requiring the coordinator to see $\Theta(\log \log n)$ distinct elements.

$(1 + \varepsilon)$**-approximation.** We note that a similar approach can be used to achieve a protocol with $O\left(\frac{\alpha}{\varepsilon^2} \cdot \log n \log \log n\right)$ total communication. Specifically, instead of stopping at a level where the coordinator sees $\Theta(1)$ unique items, the servers can choose to abort at a later time, in particular when the protocol samples down to a level where the coordinator sees $\Theta\left(\frac{1}{\varepsilon^2}\right)$ distinct elements. We show that the resulting estimator that rescales the number of distinct elements by the inverse of the sampling probability is an unbiased estimate to the number of distinct elements, and moreover that the variance is sufficiently small due to the number of samples. Hence by a standard Chebyshev argument, it follows that we can achieve a $(1 + \varepsilon)$-approximation to the number of distinct elements. We emphasize that both the constant-factor and $(1 + \varepsilon)$-approximation subsampling approach is standard among the distinct elements estimation literature, e.g., (Bar-Yossef et al., 2002; Kane et al., 2010; Woodruff & Zhang, 2014; Braverman et al., 2018; Blasiok, 2020).

However, the concern is that each of the $\alpha$ parties can send $\Omega\left(\frac{1}{\varepsilon^2}\right)$ items, resulting in $\Omega\left(\frac{\alpha}{\varepsilon^2}\right)$ items being sent across all parties. Indeed, in the hard instance of (Woodruff & Zhang, 2014), a constant fraction of items appear on a constant fraction of servers, so the protocol would actually use $\Omega\left(\frac{\alpha}{\varepsilon^2}\right)$ communication.

On the other hand, when the number of pairwise collisions is smaller than $\alpha^2 \cdot F_0(S)$, then the number of items that are redundant across multiple servers must also be smaller. We show this intuition translates to improved bounds for the same algorithm. That is, we show that when the number of pairwise collisions is $\beta \cdot F_0(S)$ for some parameter $\beta \in [1, \alpha]$, then on average, each coordinate can appear across $\sqrt{\beta}$ servers. Thus, the above protocol would send the identities of $O\left(\frac{\sqrt{\beta}}{\varepsilon^2}\right)$ items, resulting in total communication $O\left(\alpha \log n + \frac{\sqrt{\beta}}{\varepsilon^2} \log n\right)$.

Finally, we remark that when the total number of items is less than $\frac{1}{\varepsilon^2}$, i.e., $F_0(S) < \frac{1}{\varepsilon^2}$, then the same analysis suffices without any sampling at all. Moreover, since the algorithm will eventually terminate at a level where no sampling is performed if there are no previous levels with $\Theta\left(\frac{1}{\varepsilon^2}\right)$ distinct elements given to the coordinator, then the same algorithm

suffices for this case. That is, our algorithm can obliviously handle all regimes of $F_0(S)$, i.e., it does not need the promise of whether $F_0(S) \geq \frac{1}{\varepsilon^2}$ or $F_0(S) < \frac{1}{\varepsilon^2}$ as part of the input.

**Handling a smaller number of collisions.** We now describe how the guarantees of Theorem 1.1 can be further improved when the number of pairwise collisions is small. Note that $F_0(S) = F_1(S) - D$, where $D$ is the *excess mass* across all servers, which we define the excess mass of a coordinate $j \in [n]$ in a vector $v \in \mathbb{R}^n$ to be $\max(0, v_j - 1)$ and the excess mass of $v$ to be the sum of the excess masses across all of its coordinates. Note that $D$ is upper bounded by the number of pairwise collisions $C$. Thus as a simple example, if we were promised $C \leq \varepsilon F_0(S)$, then it would suffice for the $\alpha$ parties to compute $F_1(S)$, which can be done in $O(\alpha \log n)$ bits of communication.

More generally, for $C < \frac{1}{\varepsilon^2}$, it is possible to efficiently estimate $D$ without needing to send all items. In particular, given the promise that there are at most $C$ pairwise collisions, we can estimate $D$ by sampling the universe at a rate $\frac{100C}{\varepsilon^2 X^2}$, where $X$ is a constant-factor approximation to $F_0(S)$. Again by a standard expectation and variance argument, it follows that the excess mass observed by the coordinator across the items sent at this level by all players, scaled inversely by the sampling probability, is an additive $\varepsilon \cdot F_0(S)$ approximation to $D$. Since $F_0(S) = F_1(S) - D$ and the players can compute $F_1(S)$ exactly, then this provides a $(1 + \varepsilon)$-approximation to $F_0(S)$, as desired. The expected number of items sent by all items is then $O\left(\frac{C}{\varepsilon^2 \cdot F_0(S)}\right)$, which can then be translated into a concentration bound using Markov's inequality.

### A.1.2. Lower Bounds for Pairwise Collisions

We now describe our techniques for showing that the number of pairwise collisions is an inherent characteristic for the complexity of the distributed distinct elements estimation problem. We first describe our lower bound in Theorem 1.3 for a large number of pairwise collisions. We then conclude with brief intuition for our lower bound for the distributed duplication detection problem in Theorem 1.5, which immediately gives our lower bound in Theorem 1.4 for a small number of pairwise collisions.

**Large number of pairwise collisions.** The starting point for our lower bound in Theorem 1.3 is a problem called $\mathsf{SUM} - \mathsf{DISJ}$, introduced by (Woodruff & Zhang, 2014) in the coordinator model. We note that the coordinator model of communication requires messages to go from a server to the coordinator or from the coordinator to a server. Up to small factors, this can model arbitrary point-to-point communication. Indeed, if server $i$ wishes to communicate to server $j$, then server $i$ can send its message to the coordinator and have the coordinator forward it to server $j$. This increases the communication by at most a multiplicative factor of 2 and an additive $\log \alpha$ bits per message to indicate the identity of the recipient server.

In the $\mathsf{SUM} - \mathsf{DISJ}$ problem, there exist $\alpha$ players $P_1, \ldots, P_\alpha$ with inputs $X_1, \ldots, X_\alpha \in \{0,1\}^{tL}$ and a coordinator $C$ and $Y \in \{0,1\}^{tL}$. The vectors $X_1, \ldots, X_\alpha, Y$ are grouped into $t$ blocks $X_i^{(j)}, Y^{(j)}$ for $i \in [\alpha]$ and $j \in [t]$, each with $L$ coordinates. The input to each block of $X_1$ and $Y$ are first generated as an instance of two-player set disjointness, so that there are $t$ blocks of set disjointness, each with universe size $L$. Recall that on a universe of size $L$, the two-player input of set disjointness is as follows. First, for each $i \in [L]$, $i$ is given to $X_1$ with probability $\frac{1}{4}$, otherwise $i$ is given to $Y$ with probability $\frac{1}{4}$, otherwise $i$ is not given to $X_1$ or $Y$ with probability $\frac{1}{2}$. Then for a special coordinate $c$ chosen uniformly at random from $[L]$, the allocations of $c$ are reset. Then with probability $\frac{1}{2}$, $c$ is given to both players and otherwise with probability $\frac{1}{2}$, $c$ is given to neither player. The inputs $X_2, \ldots, X_\alpha$ are then generated conditioned on the value of $Y$, so that each pair $(X_i, Y)$ forms an input to two-player set disjointness. We define $\mathsf{DISJ}(X_i^{(j)}, Y^{(j)}) = 0$ if the instance is disjoint and $\mathsf{DISJ}(X_i^{(j)}, Y^{(j)}) = 1$ otherwise. The output to $\mathsf{SUM} - \mathsf{DISJ}(X_1, \ldots, X_\alpha, Y)$ is then $\sum_{i=1}^{\alpha} \sum_{j=1}^{t} \mathsf{DISJ}(X_i^{(j)}, Y^{(j)})$. (Woodruff & Zhang, 2014) show that approximating $\mathsf{SUM} - \mathsf{DISJ}(X_1, \ldots, X_\alpha, Y)$ up to additive error $O\left(\sqrt{\alpha t}\right)$ requires $\Omega(\alpha t L)$ communication.

The reduction of $F_0$ approximation from $\mathsf{SUM} - \mathsf{DISJ}$ is then as follows. Given an instance $X_1, \ldots, X_\alpha, Y$ of $\mathsf{SUM} - \mathsf{DISJ}$, the coordinator creates the indicator vector $Z$ corresponding to $[tL] \setminus Y$. It then follows that for $t = O\left(\frac{1}{\varepsilon^2 \alpha}\right)$ and $tL = \Theta\left(\frac{1}{\varepsilon^2}\right)$, a $(1 + \varepsilon)$-approximation to $F_0(X_1 + \ldots + X_\alpha + Z)$ suffices for the coordinator to determine $\mathsf{SUM} - \mathsf{DISJ}(X_1, \ldots, X_\alpha, Y)$ up to additive error $O\left(\sqrt{\alpha t}\right)$, given $Y$.

Moreover, we observe that due to the distribution of set disjointness where each coordinate is given to each player $X_i$ with probability at least $\frac{1}{4}$, then with constant probability, a constant fraction of the items are given to a constant fraction of the players. That is, with probability at least $0.99$, we have that $\Omega\left(\frac{1}{\varepsilon^2}\right)$ coordinates in the frequency vector $X_1 + \ldots + X_\alpha + Z$

have frequency $\Omega(\alpha)$.

In turns out that for $\beta < \alpha$, we can embed the same problem across $\beta$ players. Similarly, for $F_0(s) = O\left(\frac{1}{\varepsilon^2}\right)$ with $F_0(s) = \Omega(\alpha)$, it follows that for $t = O\left(\frac{F_0(s)}{\alpha}\right)$ and $tL = \Theta(F_0(s))$, a $(1+\varepsilon)$-approximation to $F_0(X_1 + \ldots + X_\alpha + Z)$ suffices for the coordinator to determine $\mathsf{SUM} - \mathsf{DISJ}(X_1, \ldots, X_\alpha, Y)$ up to additive error $O\left(\sqrt{\alpha}t\right)$, given $Y$. Putting these observations together, we obtain Theorem 1.3.

**Small number of pairwise collisions.** We first observe that our hardness result for the distributed duplication detection problem implies that any protocol that identifies whether there are fewer than $(1-\varepsilon) \cdot C$ duplicates or more than $(1+\varepsilon) \cdot C$ duplicates requires $\Omega\left(\frac{F_0(s)}{C\varepsilon^2}\right)$ communication for $C \geq \frac{4}{\varepsilon^2}$. Moreover, the hard instance of Theorem 1.5 places the $C$ pairwise collisions across unique coordinates, so that $F_0(S) = F_1(S) - C$. The $\alpha$ players can use $O\left(\log \frac{1}{\varepsilon}\right) = O(C)$ bits of communication to compute $F_1(S)$ exactly. Thus, we observe that a multiplicative $(1+\varepsilon)$-approximation to $F_0(S)$ ultimately translates to a multiplicative $\left(1 + \frac{\varepsilon \cdot F_0(S)}{C}\right)$-approximation to $C$ in the hard instance of Theorem 1.5. We then reparameterize the hardness statement to show that a multiplicative $\left(1 + \frac{\varepsilon \cdot F_0(S)}{C}\right)$-approximation to $C$ approximation requires $\Omega\left(\frac{C}{\varepsilon^2 \cdot F_0(S)}\right)$ communication. It thus remains to show Theorem 1.5, which we now describe.

### A.1.3. Distributed Duplication Detection

Our starting point for the proof of Theorem 1.5 is noting that for $C = 1$, $\varepsilon = 0$, and $\alpha = 2$, the problem becomes the decision problem of whether there exists at least a single coordinate that is duplicated or not across two sets. Thus, a natural candidate to consider is the set disjointness communication problem, e.g., (Chakrabarti et al., 2003; Bar-Yossef et al., 2004), where two players each have a subset of $[n]$ and their goal is to determine whether the intersection of their sets is empty or non-empty. See Figure 3 for an example of possible set disjointness inputs for each case. Set disjointness requires total communication $\Omega(n)$ when the sets of each player have size $\Omega(n)$, and by a simple padding argument on a smaller universe, i.e., adding dummy elements that are never included in the players' sets, it follows that $\Omega(s)$ communication is a lower bound when each set has size $\Omega(s)$.

**Handling general $\alpha$.** We first generalize to $\alpha$ players, achieving a qualitatively similar statement to that obtained via a simple reduction from set disjointness in the coordinator model, which is a problem studied in (Braverman et al., 2013). It turns out for the approximate version of the problem we will not be able to use (Braverman et al., 2013) because we will need multiple instances of a variant of promise set disjointness to argue about the number of duplicates created.

The usual notion of promise multiparty set disjointness is that there are $\alpha$ players who each have a subset of $[n]$ of size $s$ and their goal is to determine whether or not there exists a common element shared across all $\alpha$ sets. Furthermore, the generalization of existing lower bound techniques (Chakrabarti et al., 2003; Bar-Yossef et al., 2004) requires that if there is not a common element shared across all $\alpha$ sets, then the players have the promise that all of their subsets have empty pairwise intersections. That is, no element is even shared across two sets. Unfortunately for the purposes of the lower bound, there exists a simple protocol that only requires $O(s \log n)$ bits of communication: two of the $\alpha$ players simply exchange their sets (and in fact there is a more efficient way to determine if their sets are disjoint (Håstad & Wigderson, 2007)). If there is no intersection in their sets, then the intersection across all $\alpha$ sets is empty. Otherwise, if their intersection is non-empty, then by the promise of the multiparty set disjointness input, the intersection across all $\alpha$ sets must be non-empty. Thus, it would be impossible to achieve the desired $\Omega(\alpha s)$ lower bound using the usual notion of promise multiparty set disjointness.

Observe that the simple protocol results from the promise that if there is not a common element shared across all $\alpha$ sets, then the players have the promise that all of their subsets have empty pairwise intersections. For the purposes of duplication detection, this requirement is not necessary. Instead, consider a variant of multiparty set disjointness where either all $\alpha$ sets are pairwise disjoint, or there exists a single pair of sets that have non-empty intersection. In fact, we shall require our variant to be inherently distributional, uniform on each half of the universe (but not uniform on all pairs of elements of the universe), for the purposes of ultimately composing with a distributional communication problem to have an embedding from two players to $\alpha$ players while still retaining a sufficiently high entropy.

Then, intuitively, for each coordinate $j \in [n]$, the $\alpha$ parties must determine whether their input vector for $j$ is $0^\alpha$, the

elementary vector $\mathbf{e}_i$ for some $i \in [\alpha]$, or the sum of two elementary vectors $\mathbf{e}_a + \mathbf{e}_b$ for some $1 \le a < b < \alpha$. By comparison, the previous version of promise multiparty set disjointness simply required differentiating between $0^\alpha$, $\mathbf{e}_i$ for some $i \in [\alpha]$, or $1^\alpha$, which has a much larger gap, i.e., larger by a multiplicative factor of $\Omega(\alpha)$. Formally, this translates to a smaller gap by a multiplicative factor of $\Omega(\alpha)$ in the squared Hellinger distance of the protocol between the inputs of the YES and NO cases. We can then use similar arguments as in (Bar-Yossef et al., 2004) to relate the squared Hellinger distance to the information cost of a correct protocol. Our argument thus achieves tight bounds, avoiding the extraneous multiplicative factor of $\Omega(\alpha)$ overhead that would have resulted from using promise multiparty set disjointness.

**Handling general $C$ and $\varepsilon$.** Handling general $C$ and $\varepsilon$ seems significantly more challenging. A standard approach for achieving lower bounds with a $\frac{1}{\varepsilon^2}$ dependence is to utilize the Gap-Hamming communication problem (Indyk & Woodruff, 2005), in which two players Alice and Bob receive vectors $\mathbf{x}, \mathbf{y} \in \{0,1\}^t$, respectively, for $t = \Theta\left(\frac{1}{\varepsilon^2}\right)$. Their goal is to determine whether $\Delta(\mathbf{x}, \mathbf{y}) \le \frac{t}{2} - \sqrt{t}$ or $\Delta(\mathbf{x}, \mathbf{y}) \ge \frac{t}{2} + \sqrt{t}$, where $\Delta(\mathbf{x}, \mathbf{y})$ denotes the Hamming distance between $\mathbf{x}$ and $\mathbf{y}$. It is known that the Gap-Hamming problem requires $\Omega\left(\frac{1}{\varepsilon^2}\right)$ bits of communication (Chakrabarti & Regev, 2012).

Unfortunately, it is not known how to extend the Gap-Hamming problem or its variants (Pagh et al., 2014; Braverman et al., 2016) to the multiplayer communication setting. To circumvent this issue, previous works, e.g., (Woodruff & Zhang, 2012), that require Gap-Hamming-type lower bounds for multiplayer communication protocols used a composition of Gap-Hamming with a multiplayer communication problem. Although using this approach (Woodruff & Zhang, 2012) achieves a hardness of approximation for the number of distinct elements, which is the total number of items minus the number of duplicates if there are no $k$-wise collisions for $k > 2$, these results (Woodruff & Zhang, 2012; 2014) do not translate to a hardness of approximation in our case. Nevertheless, we can use the composition approach – in our case, the natural candidate is the multiplayer pairwise disjointness problem.

To that end, we define the composition problem GapSet as follows. The "outer" problem will be the GapAnd variant of Gap-Hamming while the "inner" problem will be the multiplayer pairwise disjointness problem. We first generate two vectors $\mathbf{x}, \mathbf{y} \in \{0,1\}^t$ for $t = \Theta\left(\frac{1}{\varepsilon^2}\right)$. In the YES case, we have $\langle \mathbf{x}, \mathbf{y} \rangle \ge \frac{t}{4} + c \cdot \sqrt{t}$ for some constant $c > 0$. In the NO case, we have $\langle \mathbf{x}, \mathbf{y} \rangle \ge \frac{t}{4} - c \cdot \sqrt{t}$. However, the vectors $\mathbf{x}, \mathbf{y}$ will not be given to any of the players. Using $\mathbf{x}, \mathbf{y}$, we instead generate vectors $\mathbf{u}^{(1)}, \ldots, \mathbf{u}^{(\alpha)} \in \{0,1\}^{nt}$ to give to the $\alpha$ players. Each vector $\mathbf{u}^{(i)}$ is partitioned into $t$ blocks of size $n$. For $j \in [n]$, the $j$-th block of all vectors $\mathbf{u}^{(1)}, \ldots, \mathbf{u}^{(\alpha)}$ will encode a separate instance of multiplayer pairwise disjointness. If $x_j = y_j = 1$, then there will be some "special" coordinate that is shared among two parties across the $j$-th block of the input vectors to the $\alpha$ players, i.e., $j$-th instance of multiplayer pairwise disjointness. Otherwise, if $x_j = 0$ or $y_j = 0$ (or both), then no coordinate is shared among multiple parties in the $j$-th instance of multiplayer pairwise disjointness. The goal of the $\alpha$ players is to distinguish whether (1) there exist at least $\frac{t}{4} + c \cdot \sqrt{t}$ blocks that have some pairwise intersection, or (2) there are at most $\frac{t}{4} + c \cdot \sqrt{t}$ blocks that have some pairwise intersection.

Intuitively, the GapAnd lower bounds show that $\Omega(t)$ communication is needed to solve GapAnd. However, to recover each coordinate of $\mathbf{x}$ and $\mathbf{y}$, the players must essentially solve an instance of multiplayer pairwise disjointness, which requires $\Omega(n)$ communication, leading to a lower bound of $\Omega(nt)$ overall communication for the GapSet problem.

The reduction to duplication detection is then relatively straightforward. For $C = O\left(\frac{1}{\varepsilon^2}\right)$, we choose $t$ to be $\Theta(C)$ and $n$ to be $\Omega\left(\frac{\alpha s}{C}\right)$, so that $\Omega(nt) = \Omega(\alpha s)$. For the case where $C = \Omega\left(\frac{1}{\varepsilon^2}\right)$, we instead set $t = \frac{1}{\varepsilon^2}$ and $n = \Omega\left(\frac{\alpha s}{C}\right)$, which gives the desired $\Omega(nt) = \Omega\left(\frac{\alpha s}{C \varepsilon^2}\right)$ communication lower bound. However, the number of duplicates in this distribution is not correct. Thus, we copy the instance $\Theta\left(\frac{C}{t}\right)$ times to account for this. It then remains to prove the hardness of the GapSet problem.

**Hardness of GapSet.** A natural approach for showing that the information cost of multiple instances of a communication problem is the sum of the information costs of each instance of the communication problem, is the direct sum approach. The approach generally proceeds by embedding multiple independent instances of the inner problem into coordinates of an outer problem to show the hardness of the composition. For example, (Bar-Yossef et al., 2004) views set disjointness as the $n$-wise OR of the AND problem across $\alpha$ players and uses the direct sum framework to show that since each AND problem requires $\Omega(1)$ information, then set disjointness requires $\Omega(n)$ information. However, the direct sum approach is generally used for composition problems where the outer problem is sensitive to changes in the input – in the example above, the OR problem has different values for $0^n$ and for any elementary vector $\mathbf{e}_j$ with $j \in [n]$. Our outer problem GapAnd does not necessarily satisfy this condition and so it does not seem that we can apply a direct sum argument.

Instead, we take the reverse approach by starting with the outer problem in the composition and using it to solve the inner

problem, which is an approach used in (Woodruff & Zhang, 2012) but for a very different multiplayer communication problem.

For each $j \in [t]$, we let $D_j = |x_j \cap y_j|$, so that the GapAnd problem is simply to determine whether $|D_j| \geq \frac{t}{4} + c \cdot \sqrt{t}$ or $|D_j| \leq \frac{t}{4} - c \cdot \sqrt{t}$. It is known that any protocol $\Pi$ that solves GapAnd must reveal $\Omega(t)$ information about $D_1, \ldots, D_t$. In particular, this implies that there exist $\Omega(t)$ coordinates $j \in [t]$ such that conditioned on the previous values $D_1, \ldots, D_{j-1}$, the protocol reveals $\Omega(1)$ information about $D_j$; we call such a coordinate an *informative* index. The goal is then to show that for any informative index, the protocol $\Pi$ must reveal $\Omega(n)$ information, due to the hardness of the multiplayer pairwise disjointness problem.

Consider a hard-wiring of a specific instance $\mathbf{V}$ of the multiplayer pairwise disjointness problem on an informative index $j \in [t]$. The $\alpha$ players can fix the other coordinates of the GapSet before the $j$-th block and then plant $\mathbf{V}$ on the $j$-th block. Because the protocol reveals $\Omega(1)$ information about $D_j$, this translates to an $\Omega(1)$ additive advantage over random guessing by using the maximum likelihood estimator. Moreover, note that the conditioning of the special coordinate in the multiplayer pairwise disjointness can only change the conditional information cost by an additive logarithmic factor. Hence, we obtain a protocol $\Pi'$ that can be used to solve multiplayer pairwise disjointness, which requires $\Omega(n)$ information. Then summing over the $\Omega(t)$ informative indices, it follows that the information cost of $\Pi$ for GapSet is $\Omega(nt)$.

## A.2. Extension to Streaming Algorithms

Motivated by the connection between distributed algorithms and streaming algorithms, we also consider parameterized algorithms for the distinct elements problem in the streaming setting. In the streaming setting, a low memory algorithm is given a single pass or a small number of passes over a stream of items, and should output a $(1 \pm \varepsilon)$-approximation to the number $F_0$ of distinct items at the end of the stream with constant probability. We consider insertion-only streams.

We choose to parameterize the complexity in terms of $C$, the number of coordinates $i \in [n]$ with frequency $f_i > 1$. In situations in which the data is Zipfian, it may be the case that $C$ is small compared to $F_0$, as only a few items may occur more than once. We note that without parameterizing by $C$, there is an $\Omega\left(\frac{1}{\varepsilon^2} + \log n\right)$ bit lower bound for any constant number of passes (Jayram & Woodruff, 2013). We, however, are able to give a two-pass streaming algorithm using $O(C + 1/\varepsilon)$ memory, up to logarithmic factors, and we also prove a matching $\Omega\left(C + \frac{1}{\varepsilon}\right)$ lower bound for any constant number of passes. For one-pass algorithms we are able to achieve $O\left(\frac{C}{\varepsilon}\right)$ bits of space, up to logarithmic factors, also showing that one can bypass the $\Omega\left(\frac{1}{\varepsilon^2}\right)$ lower bound for $C < 1/\varepsilon$. In fact, in all of our results, we can replace $C$ with the number of items with frequency strictly larger than 1, times the minimum of 1 and $\frac{1}{\varepsilon^2 F_0}$, which is especially useful if $F_0 \gg \frac{1}{\varepsilon^2}$.

Our two-pass streaming algorithm is based on computing the $\ell_1$-norm of the underlying vector, which is just the stream length since we only consider insertions of items, and then subtracting off an estimate to the contribution of items with frequency strictly larger than 1, which we call *outliers*. We then add back in an estimate to the total number of outliers. We can assume $F_0 = \Theta\left(\frac{1}{\varepsilon^2}\right)$ by subsampling the universe items at $O(\log n)$ scales to reduce $F_0$ to value in $\Theta\left(\frac{1}{\varepsilon^2}\right)$ and preserve its value up to $1 \pm \varepsilon$ (for this discussion let us assume $F_0$ is at least $1/\varepsilon^2$ to begin with). We observe that for items that have frequency smaller than $\frac{1}{\varepsilon C}$, their total contribution to the $\ell_1$-norm is $\frac{1}{\varepsilon}$, so they can be ignored as $F_0 = \Theta\left(\frac{1}{\varepsilon^2}\right)$. For items with frequency sufficiently large, we can identify them all using a CountSketch data structure (Charikar et al., 2002) with $O\left(C + \frac{1}{\varepsilon}\right)$ buckets, and thus this amount of memory up to logarithmic factors. Further, we can obtain an unbiased estimate to their sum with small enough variance by adding their individual CountSketch estimates. Finally, there are items with "intermediate frequencies" for which we cannot find them with CountSketch - for these we instead subsample the universe elements, making the surviving universe elements with intermediate frequencies "heavier", since the total $F_0$ has gone down and thus the noise in each CountSketch bucket is smaller while the frequency of an item with intermediate frequency has remained the same (note that we subsample universe elements rather than stream items). We identify the surviving universe elements with intermediate frequency and scale back up by the inverse of the sampling probability to estimate their contribution to the $\ell_1$-norm.

One issue is that for items with intermediate frequencies, we need to subsample at multiple geometric rates, and in order to avoid over-counting we need to learn their frequency counts exactly, which we accomplish with a second pass. However, if we were to instead increase the number of hash buckets of CountSketch from $O\left(C + \frac{1}{\varepsilon}\right)$ to $O\left(\frac{C}{\varepsilon}\right)$, we could find all heavy hitters in a single pass. Interestingly, for our single pass algorithm, we can also use methods for robust mean estimation (Prasad et al., 2019) applied to the values of our CountSketch buckets to estimate $F_0$. Indeed, we can view the few CountSketch buckets which contain an outlier as the corrupted samples in a robust mean estimation algorithm. This

allows us to save a $\log n$ factor from the number of CountSketch tables. We give these results in Appendix D.

## B. Communication Game Lower Bound

In this section, we define and analyze the communication complexity of the GapSet problem. Recall that GapSet is the composition of the GapAnd problem on $t$ coordinates with the multiplayer pairwise disjointness problem on $n$ coordinates. The intuition is that GapSet requires $\Omega(nt)$ communication because the outer problem GapAnd requires $\Omega(t)$ communication and each inner problem of the multiplayer pairwise disjointness problem requires $\Omega(n)$ communication. The formal proof is significantly more involved. For starters, communication complexity is not additive but information costs are, so we require a number of preliminaries for information theory, which we recall in part in Section A.

We first formally define the GapAnd problem, which serves as the "outer" problem in the composition problem GapSet, as follows:

**Definition B.1** (GapAnd). *In the distributional $t$-coordinate GapAnd problem $\mathsf{GapAnd}_t$, Alice and Bob receive vectors $\mathbf{x}, \mathbf{y} \in \{0, 1\}^t$ generated uniformly at random. Let $c > 0$ be a constant. Bob's goal is to determine whether the input falls into the cases:*

- *In the YES case, for at least $\frac{t}{4} + \frac{1}{16} \cdot \sqrt{t}$ coordinates $j \in [t]$, we have $x_j = y_j = 1$.*

- *In the NO case, for at most $\frac{t}{4} - \frac{1}{16} \cdot \sqrt{t}$ coordinates $j \in [t]$, we have $x_j = y_j = 1$.*

- *Otherwise if neither the YES case nor the NO case occurs, then Bob's output may be arbitrary.*

It is known that any protocol that obtains a constant advantage over random guessing reveals $\Omega(t)$ information about the input vectors.

**Lemma B.2.** *(Chakrabarti et al., 2012; Pagh et al., 2014; Braverman et al., 2016) There exists a sufficiently small constant $\delta > 0$ for which any private randomness protocol $\Pi$ for $\mathsf{GapAnd}_t$ that succeeds with probability at least $\frac{1}{2} + \Omega(1)$ over inputs $\mathbf{x}, \mathbf{y}$, the private randomness of $\Pi$ and public randomness $R$ satisfies*

$$I(\Pi(\mathbf{x}, \mathbf{y}); \mathbf{x}, \mathbf{y} | R) = \Omega(t).$$

Recall that in the set disjointness communication problem, e.g., (Chakrabarti et al., 2003; Bar-Yossef et al., 2004), two players each have a subset of $[n]$ and their goal is to determine whether the intersection of their sets is empty or non-empty. We define a variant of set disjointness as our "inner" problem. See Figure 3 for an example of possible set disjointness inputs for each case.

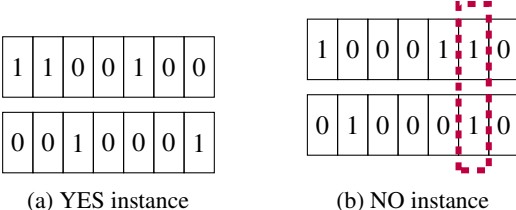

(a) YES instance        (b) NO instance

Figure 3: Examples of YES and NO instances of the set disjointness problem for $\alpha = 2$ players.

We show that any protocol for $\mathsf{GapSet}_{n,k}$ with constant $k$ requires $\Omega(n)$ communication. To that end, we first recall the following structural properties from (Bar-Yossef et al., 2004):

**Lemma B.3** (Lemma 6.2 in (Bar-Yossef et al., 2004)). *Let $f(X)$ and $f(Y)$ be two random variables and let $Z$ be a random variable with uniform distribution in $\{X, Y\}$. If $Z$ is independent of both $f(X)$ and $f(Y)$, then $I(Z; f(Z)) \geq h^2(f_X, f_Y)$, where $f_X$ denotes the distribution of $f$ on $X$.*

**Lemma B.4** (Cut-and-Paste, e.g., Lemma 6.3 in (Bar-Yossef et al., 2004)). *Let $\Pi$ be a randomized protocol, $x, x' \in X$, and $y, y' \in Y$ for some domains $X, Y$. Then $h(\Pi_{xy}, \Pi_{x'y'}) = h(\Pi_{xy'}, \Pi_{x'y})$.*

**Lemma B.5** (Pythagorean Lemma, e.g., Lemma 6.4 in (Bar-Yossef et al., 2004)). *Let $\Pi$ be a randomized protocol, $x, x' \in X$, and $y, y' \in Y$ for some domains $X, Y$. Then $h^2(\Pi_{xy}, \Pi_{x'y}) + h^2(\Pi_{xy'}, \Pi_{x'y'}) \leq 2h^2(\Pi_{xy}, \Pi_{x'y'})$.*

**Lemma B.6** (Lemma 6.5 in (Bar-Yossef et al., 2004))**.** *Let $\Pi$ be a randomized protocol with failure probability $\delta \in (0, 1)$ for a function $f$. Then for any two input pairs $(x, y)$ and $(x', y')$ with $f(x, y) \neq f(x', y')$, we have $h^2(\Pi_{xy}, \Pi_{x'y'}) \geq 1 - 2\sqrt{\delta}$.*

**Lemma B.7.** *Let $\alpha \geq 2$ be an integer and let $k \in [\alpha]$. Let $\Pi$ be a randomized $\alpha$-party communication protocol with inputs from $\{0, 1\}^\alpha$ for determining whether exactly $\ell$ coordinates are one and let $\pi : [\alpha] \to [\alpha]$ be any permutation. Then,*

$$\sum_{i=1}^{\alpha} h^2(\Pi_{0^\alpha}, \Pi_{\mathbf{e}_i}) \geq \frac{1}{2k} \sum_{j=1}^{\lfloor \alpha/k \rfloor} h^2(\Pi_{0^\alpha}, \Pi_{I_j}),$$

*where $I_j = \left[ \frac{\alpha(j-1)}{2k} + 1, \frac{\alpha j}{2k} \right]$.*

*Proof.* Similar to the proof of Lemma 7.2 in (Bar-Yossef et al., 2004), we prove the claim by using an induction argument on a tree. Let $\alpha'$ be the smallest power of two such that $\alpha \leq \alpha'$ and let $\phi$ be any mapping that extends a permutation $\pi : [\alpha] \to [\alpha]$ in the natural way to $\alpha'$ coordinates, i.e., $\phi(i) = \pi(i)$ for $i \in [\alpha]$ and $\phi(i) = i$ for $i \in (\alpha, \alpha']$. Let $T$ be a complete binary tree of height $\log(\alpha')$ with leaves labeled from $\phi(1)$ to $\phi(\alpha')$ and internal nodes labeled with the leaves in their corresponding rooted subtrees.

For $a, b \in [\alpha]$ with $b - a + 1 = k$, let $c = \left\lfloor \frac{a+b}{2} \right\rfloor$. Let $u = \mathbf{e}_{\pi([a,b])}$, $v = \mathbf{e}_{\pi([a,c])}$, and $w = \mathbf{e}_{\pi([c+1,b])}$. By an analogue of Lemma B.4 for $\alpha'$-player communication games, we have that

$$h(\Pi_{0^{\alpha'}}, \Pi_u) = h(\Pi_v, \Pi_w).$$

On the other hand, since the last $\alpha' - \alpha$ input coordinates are known by all players, we have

$$h(\Pi_{0^\alpha}, \Pi_u) = h(\Pi_v, \Pi_w).$$

By the triangle inequality,

$$h(\Pi_{0^\alpha}, \Pi_v) + h(\Pi_{0^\alpha}, \Pi_w) \geq h(\Pi_v, \Pi_w).$$

Thus by the Cauchy-Schwarz inequality,

$$h^2(\Pi_v, \Pi_w) \leq 2(h^2(\Pi_{0^\alpha}, \Pi_v) + h^2(\Pi_{0^\alpha}, \Pi_w)).$$

Then by induction,

$$2(b - a + 1) \sum_{i=a}^{b} h^2(\Pi_{0^\alpha}, \Pi_{\mathbf{e}_i}) \geq h^2(\Pi_{0^\alpha}, \Pi_{\mathbf{e}_{[a,b]}}).$$

For $k = b - a + 1$, we can split the interval $[0, \alpha]$ into at least $\left\lfloor \frac{\alpha}{k} \right\rfloor$ disjoint intervals of length $k$. Therefore,

$$\sum_{i=1}^{\alpha} h^2(\Pi_{0^\alpha}, \Pi_{\mathbf{e}_i}) \geq \frac{1}{2k} \sum_{j=1}^{\lfloor \alpha/k \rfloor} h^2(\Pi_{0^\alpha}, \Pi_{I_j}),$$

where $I_j = \left[ \frac{\alpha(j-1)}{2k} + 1, \frac{\alpha j}{2k} \right]$. □

We now show that any protocol for $\mathsf{GapSet}_{n,k}$ requires $\Omega(n)$ communication for constant $k > 0$.

**Lemma B.8.** *The conditional information cost of any algorithm for $\mathsf{SetDisj}_{n,k}$ that succeeds with probability $\frac{1}{2} + \Omega(1)$ is $\Omega(n)$.*

*Proof.* We first use the direct sum paradigm by defining the NO distribution $\zeta$ for the single coordinate $\mathsf{SetWt}_{n,k}$ problem, which implicitly forms the NO distribution $\mu_0$ for $\mathsf{SetDisj}_{n,k}$. We use a random variable $Q \in [\alpha]$ chosen uniformly at random, so that conditioned on the value of $Q = i \in [\alpha]$, we have that $\mathbf{u}$ is chosen from $\{0^\alpha, \mathbf{e}_i\}$ uniformly at random, implicitly defining the distribution $\zeta$. Then $\zeta^n$ is the collapsing distribution that matches the distribution of $\mu_0$ and so by Theorem A.14, it suffices to show that the conditional information cost of $\Pi$ with respect to $\zeta$ is $\Omega(1)$.

To that end, we have

$$I(\Pi(\mathbf{u}); \mathbf{u}|Q) = \frac{1}{\alpha}\sum_{i=1}^{\alpha} I(\Pi(\mathbf{u}); \mathbf{u}|Q = i).$$

By Lemma B.3 and Lemma B.7

$$I(\Pi(\mathbf{u}); \mathbf{u}|Q) \geq \frac{1}{\alpha}\sum_{i=1}^{\alpha} h^2(\Pi_{0^{\alpha}}, \Pi_{\mathbf{e}_i})$$

$$\geq \frac{1}{2\alpha}\sum_{j=1}^{\lfloor \alpha/k \rfloor} h^2(\Pi_{0^{\alpha}}, \Pi_{\mathbf{e}_{2j-1}+\mathbf{e}_{2j}}),$$

By the correctness of the protocol on $\mu$ and Lemma B.6, we have that $I(\Pi(\mathbf{u}); \mathbf{u}|Q) = \Omega(1)$. Hence, the conditional information cost of any algorithm for $\mathsf{SetDisj}_{n,k}$ that succeeds with probability $\frac{1}{2} + \Omega(1)$ under the mixture distribution $\mu = \frac{1}{2}\mu_0 + \frac{1}{2}\mu_1$ is $\Omega(n)$. $\qquad\square$

We now define the composition communication problem GapSet.

**Definition B.9** (GapSet). *In $\mathsf{GapSet}_{t,\alpha,n,k}$, the distribution $\phi$ for the GapSet problem over $t$ blocks each with $n$ coordinates and $k$ collisions for $\alpha$ players is defined as follows. The $\alpha$ players receive input vectors $\mathbf{u}^{(1)}, \ldots, \mathbf{u}^{(\alpha)} \in \{0,1\}^{nt}$ such that the $i$-th player receives vector $\mathbf{u}^{(i)}$, for all $i \in [\alpha]$. Let $\beta = \lfloor \frac{\alpha}{2} \rfloor$ and $\ell = \lfloor \frac{k}{2} \rfloor$. The input is generated by first drawing $\mathbf{x}, \mathbf{y} \in \{0,1\}^t$ from $\mathsf{GapAnd}_t$ and creating input vectors $\{\mathbf{u}^{(i,j)}\}_{i\in[\alpha],j\in[t]}$.*

- *For each $j \in [t]$, a special coordinate $Z_j \in [n]$ is selected uniformly at random.*
  - *If $x_j = 1$, then the coordinate $Z_j$ is given to $\ell$ random players, i.e., the protocol selects $\ell$ indices that are at most $\beta$, i.e., $i_{j_1}, \ldots, i_{j_\ell} \leq \beta$, and sets $u_{Z_j}^{(i_{j_1},j)} = \ldots = u_{Z_j}^{(i_{j_\ell},j)} = 1$.*
  - *If $y_j = 1$, then the coordinate $Z_j$ is given to $k - \ell$ random players, i.e., the protocol selects $k - \ell$ indices that are at least $\beta + 1$, i.e., $i_{j_{\ell+1}}, \ldots, i_{j_k} \geq \beta + 1$, and sets $u_{Z_j}^{(i_{j_{\ell+1}},j)} = \ldots = u_{Z_j}^{(i_{j_k},j)} = 1$.*

- *For all coordinates $w \in [n]$ with $w \neq Z_j$, with probability $\frac{1}{2}$, the protocol assigns $w$ to a random player $w_j \in [\alpha]$, i.e., it sets $u_w^{(w_j,j)} = 1$.*

*Each vector $\mathbf{u}^{(j)} = \mathbf{u}^{(1,j)} \circ \mathbf{u}^{(2,j)} \circ \ldots \circ \mathbf{u}^{(\alpha,j)}$ and the $\alpha$ players' goal to determine whether the input falls into the cases:*

- *In the YES case, the input is generated from vectors $\mathbf{x}, \mathbf{y} \in \{0,1\}^t$ that are in the YES case for $\mathsf{GapAnd}_t$.*

- *In the NO case, the input is generated from vectors $\mathbf{x}, \mathbf{y} \in \{0,1\}^t$ that are in the NO case for $\mathsf{GapAnd}_t$.*

- *Otherwise if the input is generated from vectors $\mathbf{x}, \mathbf{y} \in \{0,1\}^t$ that are neither in the YES or NO cases for $\mathsf{GapAnd}_t$, then the $\alpha$ players' output may be arbitrary.*

See Figure 4 for examples of possible inputs to GapSet.

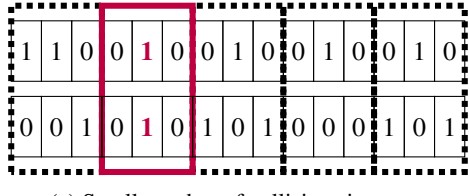

(a) Small number of collisions instance

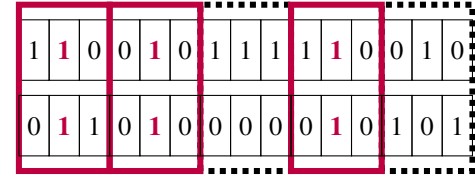

(b) Large number of collisions instance

Figure 4: Examples of input instances for GapSet problem for $\alpha = 2$ players, $t = 5$ blocks, and $n = 3$ coordinates on each block.

We first show the mutual information between a successful protocol for GapSet and a set of auxiliary variables.

**Lemma B.10.** *Let $\Pi$ be a protocol that solves* $\mathsf{GapSet}_{t,\alpha,n,k}$ *with probability at least* $0.99$. *Let* $\{\mathbf{u}^{(i)}\}_{i\in[\alpha]}$ *be an input for* $\mathsf{GapSet}_{t,\alpha,n,k}$, *generated from vectors* $\mathbf{x}, \mathbf{y}$ *drawn from* $\mathsf{GapAnd}_t$. *Let $M$ be the transcript of $\Pi$ on input* $\{\mathbf{u}^{(i)}\}_{i\in[\alpha]}$ *and $R$ be a fixing of auxiliary random bits. For each $j \in [t]$, let $D_j = |x_j \cap y_j|$. Then*

$$I(M; D_1, \ldots, D_t | R) = \Omega(t).$$

*Proof.* Since the vectors $\mathbf{x}, \mathbf{y}$ are an instance drawn from $\mathsf{GapAnd}_t$, then we have $x_j = y_j = 1$ if and only if $D_j = 1$. Thus $M$ any message produced by a protocol $\Pi$ that solves the distributional problem $\mathsf{GapSet}_{t,\alpha,n,k}$ with probability at least $0.99$ also solves $\mathsf{GapAnd}_t$ with probability at least $0.99$,

By Lemma B.2,

$$I(M; D_1, \ldots, D_t | R) \geq I(\Pi(\mathbf{A}, \mathbf{B}); \mathbf{A}, \mathbf{B} | R) = \Omega(t).$$

$\square$

We define the following "guess" variant of the set disjointness problem, along with the input distributions $\mathcal{D}_1$ and $\mathcal{D}_2$. In the GUESS problem, there exists a fixed coordinate $Z \in [n]$. We define the distribution $D = (D_1, \ldots, D_n)$ as follows. For each $i \in [n]$, $D_i$ is a random integer in $[\alpha]$. All sites not equal to $D_i$ have their $i$-th coordinate set to zero. With probability $\frac{1}{2}$, the site $D_i$ has its $i$-th coordinate set to one; otherwise it is set to zero. We define $\mathcal{D}_1$ to be this distribution.

We then achieve the distribution $\mathcal{D}_2$ by making the following modifications to $\mathcal{D}_1$. For coordinate $Z$, with probability $\frac{1}{2}$, we set the $Z$-th coordinate of $\ell = \lfloor \frac{k}{2} \rfloor$ random servers of index at most $\beta = \lfloor \frac{\alpha}{2} \rfloor$ to one and the remaining to zero. In this case, we say $X = 1$. Otherwise, we set all of those coordinates to be zero and we say $X = 0$. Similarly for coordinate $Z$, with probability $\frac{1}{2}$, we set the $Z$-th coordinate of $k - \ell$ random servers of index larger $\beta = \lfloor \frac{\alpha}{2} \rfloor$ to one and the remaining to zero. In this case, we say $Y = 1$. Otherwise, we set all of those coordinates to be zero and we say $Y = 0$. In the GUESS variant, there is an additional party that observes the transcript of the communication protocol between Alice and Bob and must guess the values of $X$ and $Y$. By a similar argument to Theorem 5 in (Woodruff & Zhang, 2012), we have:

**Theorem B.11.** *Let $\Pi$ be the transcript of any randomized protocol for GUESS on input $V \sim \mathcal{D}_2$ with success probability $\frac{1}{4} + \Omega(1)$. Then for $k = \Theta(1)$, we have $I(V; \Pi \mid D, Z) = \Omega(n)$, where information is measured with respect to $\mathcal{D}_2$.*

The only difference is that the input distribution $\mathcal{D}_2$ is slightly different than the input distribution for Theorem 5 in (Woodruff & Zhang, 2012), where all $\alpha$ servers have ones in the case the set disjointness input is a NO instance. By comparison, we only have $k$ servers for $k = O(1)$, so that the mutual information is $\Omega(1)$ times the Hellinger distance between the all zeros vector and an elementary vector.

We now lower bound the mutual information for any protocol that solves $\mathsf{GapSet}$. The proof follows exactly the same structure as Theorem 7 of (Woodruff & Zhang, 2012).

**Lemma B.12.** *Let $R$ be a source of fixed randomness and $\mathbf{U} = \{\mathbf{u}^{(i)}\}_{i\in[\alpha]}$ be an instance of $\mathsf{GapSet}_{t,\alpha,n,k}$. For each $j \in [t]$, let $D^{(j)}$ and $Z^{(j)}$ denote the outcomes of $D$ and $Z$ for the $j$-th block, respectively and let $\vec{D}$ and $\vec{Z}$ denote the outcomes across all $j \in [t]$. Then for any protocol $\Pi$ that produces transcript $M(\mathbf{U})$ on input $\mathbf{U}$ that solves $\mathsf{GapSet}_{t,\alpha,n,k}$ with probability at least $\frac{3}{4} + \Omega(1)$ satisfies $I(M(\mathbf{U}); \mathbf{U} | \vec{D}, \vec{Z}) \geq \Omega(nt)$.*

*Proof.* Let $\{\mathbf{u}^{(i)}\}_{i\in[\alpha]}$ be generated from vectors $\mathbf{x}, \mathbf{y}$ drawn from $\mathsf{GapAnd}_t$. For each $j \in [t]$, let $X^{(j)}$ and $Y^{(j)}$ denote the outcomes of $X$ and $Y$ for the $j$-th block, respectively, and define $\vec{X}$ and $\vec{Y}$ similarly. By Lemma B.10 and the chain rule, i.e., Fact A.9, there exist $\Omega(t)$ coordinates $j \in [t]$ such that $I(X^{(j)}, Y^{(j)}; M \mid D^{(<j)}, X^{(<j)}, Y^{(<j)}, Z^{(<j)}) = \Omega(1)$. We define such an index $j \in [t]$ to be *informative*.

Now, we can write $I(\mathbf{U}^{(<j)}; M \mid D, X^{(<j)}, Y^{(<j)}, Z)$ as

$$\sum_{(d,x,y,z)} \mathbf{Pr}\left[(D^{(<j)}, X^{(<j)}, Y^{(<j)}, Z^{(<j)}) = (d,x,y,z)\right]$$
$$\times I(\mathbf{U}^{(<j)}; M \mid D^j, Z^j, (D^{(<j)}, X^{(<j)}, Y^{(<j)}, Z^{(<j)}) = (d,x,y,z)).$$

By definition of an informative index $i$, we have with constant fraction of the summands $(d, x, y, z)$ that

$$I(\mathbf{U}^{(<j)}; M \mid D^j, Z^j, (D^{(<j)}, X^{(<j)}, Y^{(<j)}, Z^{(<j)}) = (d,x,y,z)) = \Omega(1).$$

We say such ordered tuplets are *good* for an informative index $j$.

Observe that by independence of the distribution across the coordinates $j \in [n]$, we have

$$H(\mathbf{U}^{(<j)} \mid D^j, Z^j, (D^{(<j)}, X^{(<j)}, Y^{(<j)}, Z^{(<j)}) = (d, x, y, z)) = 2.$$

Now for an informative index $j$ and $(d, x, y, z)$ that is good for $j$,

$$\begin{aligned}
&H(\mathbf{U}^{(<j)} \mid M, D^j, Z^j, (D^{(<j)}, X^{(<j)}, Y^{(<j)}, Z^{(<j)}) = (d, x, y, z)) \\
&= H(\mathbf{U}^{(<j)} \mid D^j, Z^j, (D^{(<j)}, X^{(<j)}, Y^{(<j)}, Z^{(<j)}) = (d, x, y, z)) \\
&\quad - I(\mathbf{U}^{(<j)}; M \mid D^j, Z^j, (D^{(<j)}, X^{(<j)}, Y^{(<j)}, Z^{(<j)}) = (d, x, y, z)) \\
&= 2 - \Omega(1).
\end{aligned}$$

For a good $(d, x, y, z)$ for an informative $j$, we build a protocol $\Pi_{j,d,x,y,z}$ that computes the GUESS problem on input $\mathcal{D}_2$ with probability $\frac{1}{4} + \Omega(1)$. Let $A_1, \ldots, A_\alpha$ be the inputs to the $\alpha$ sites and let $\{Q, R\}$ be the input for the predictor, where $Q$ is a realization of $D$ and $R$ is a realization of $Z$. Then $\Pi_{j,d,x,y,z}$ has $(j, d, x, y, z)$ hard-coded so that the $k$ sites construct an input $B$ for GapSet using the distribution $\phi$. In particular, they set $B^{(j)} = \{A_1, \ldots, A_\alpha\}$ and then use private randomness to independently generate blocks $j' \neq j$, given the values of $(d, x, y, z)$. Then by setting $D^{(j)} = Q$ and $Z^{(j)} = R$ for the predictor, the sites can run $\Pi$ on input $B$.

Hence by Theorem B.11, we have

$$\begin{aligned}
I(M(\mathbf{U}); \mathbf{U} \mid \vec{D}, \vec{Z}) &\geq \sum_{\text{informative } j} I(M(\mathbf{U}); \mathbf{U}^{(j)} \mid \vec{D}, \vec{Z}, \mathbf{U}^{(<j)}) \\
&\geq \sum_{\text{informative } j} \sum_{\text{good } (d,x,y,z)} \mathbf{Pr}_{(d,x,y,z)} \left[ (D^{(<j)}, X^{(<j)}, Y^{(<j)}, Z^{(<j)}) = (d, x, y, z) \right] \\
&\quad \times I(\mathbf{U}^{(<j)}; M \mid D^j, Z^j, (D^{(<j)}, X^{(<j)}, Y^{(<j)}, Z^{(<j)}) = (d, x, y, z)) \\
&= \Omega(t) \cdot \Omega(n) = \Omega(nt).
\end{aligned}$$

$\square$

Finally, we show the communication complexity of GapSet.

**Theorem B.13.** *Any protocol $\Pi$ that solves* $\mathsf{GapSet}_{t,\alpha,n,k}$ *with probability at least $\frac{3}{4} + \Omega(1)$ uses $\Omega(nt)$ bits of communication.*

*Proof.* The proof follows immediately from Lemma B.12, Fact A.5, and Lemma A.8. $\square$

Using Lemma B.12, we ultimately show Theorem 1.5 by designing a reduction with the appropriate values of $n$ and $t$; we defer the proof to Section C.

## C. Applications to Duplication Detection

In this section, we prove Theorem 1.5 as well as describe additional applications to the duplication detection problem. Recall that in the duplication detect problem, the goal for the $\alpha$ servers is to approximate the total number of duplicates, where a duplicate is defined to be a coordinate $j \in [n]$ that appears on at least two distinct servers. We first show that our communication game lower bound in Section B gives a hardness of approximation to estimating the number of duplicates. We then give an algorithm that shows that our lower bound is in fact, tight.

### C.1. Motivation and Related Work

**Database cleaning.** The detection and elimination of repeated data is an important task for data cleaning and data quality in database management. Due to data entry errors, multiple accounts, varying conventions, or a number of other reasons, the same concept or real-world entity may correspond to multiple entries in a database, which can often be the source of

significant challenges for users of the database. For example, duplicated entries in a database can lead to increased direct mailing costs because duplicated consumers may be sent multiple copies of the same catalog. (Chaudhuri et al., 2005) observes that duplicates can also induce incorrect outputs for analytic queries, such as the size of the overall consumer base, and thus lead to erroneous data mining models downstream. Therefore, significant effort and costs are spent on identifying and removing repeated items in a database.

The duplication detection problem is also known as the merge/purge, deduping, and record linkage problem (Fellegi & Sunter, 1969; Kilss & Alvey, 1986; Bitton & DeWitt, 1983; Monge & Elkan, 1997; Hernández & Stolfo, 1998; Sarawagi & Bhamidipaty, 2002; Bilenko & Mooney, 2003) and in fact, a more general version of the duplication detection problem, known as fuzzy duplication detection (Ananthakrishna et al., 2002; Chaudhuri et al., 2005), is often studied. In the fuzzy duplication problem, the goal is not only to identify the duplicated entries in a database, but also to identify sufficiently close entries in a database, which among other things can help account for human errors such as typos or missing information.

Techniques for the fuzzy duplication problem can largely be separated into supervised and unsupervised approaches. Supervised approaches generally use training data containing known duplicates to find characteristics that are then used to identify future duplicates (Cohen & Richman, 2002; Bilenko & Mooney, 2003). However, such approaches not only lack theoretical guarantees but also often encounter difficulties in finding effective training data that display the same distribution of duplicates observed in practice (Chaudhuri et al., 2005). (Sarawagi & Bhamidipaty, 2002; Tejada et al., 2002) addressed this shortcoming by using active learning to interactively update the training data, but the drawback is that their approaches require manual guidance to perform the updates. Unsupervised approaches generally use global thresholds for distance functions, such as the edit distance or the cosine distance, to detect duplicates, which induces poor recall-precision tradeoffs (Hernández & Stolfo, 1998; Monge & Elkan, 1997; Chaudhuri et al., 2005) since individual fuzzy duplications may not follow the general global trends. For the standard duplication detection problem, one simple approach (Metwally et al., 2005) is to use Bloom filters (Bloom, 1970) to avoid communicating the full information of each entry. However, this generally requires communication linear in the size of the entire dataset.

**Advertising commissions.** As noted by (Metwally et al., 2005), a related application of duplicate detection is to the task of advertising commissions by the middle party between advertisers and publishers. In a typical scenario, an advertiser and a publisher generally arrange an agreement for a specified commission for each user interaction with an advertisement, such as clicking or watching an advertisement, completing a form, bidding on an item, or making a transaction. The publisher will then display advertisements, forms, or text links on its products, while using tracking information, e.g., referral codes, to monitor the traffic it directs to the advertiser's materials. Similarly, the advertiser will also use tracking information to monitor the traffic directed from each publisher to its site. Inconsistencies between the statistics claimed by the advertiser and the publisher are then resolved by an arbiter, a third-party tracking entity called the advertising commissioner, who serves as a middle party who receives the traffic from the publisher and forwards it to the advertiser.

Because the publisher receives revenue proportional to the traffic that they direct to the advertising site, they can be incentivized to falsely increase the number of hits produced by their referral link, a process called click inflation (Anupam et al., 1999; Metwally et al., 2005). On the other hand, the advertising commissioner must report the true traffic directed to the advertising site and thus detect any click inflation that has occurred. By simplifying the traffic information to an identification number such as the IP address, the problem for the advertising commissioner reduces to duplication detection.

**Duplication detection in the streaming model.** Duplication detection has also been extensively studied in other big data models, in particular the streaming model (Metwally et al., 2005; Gopalan & Radhakrishnan, 2009; Jowhari et al., 2011). In fact, (Muthukrishnan, 2005) explicitly ask the question of whether duplication detection could be performed in $\text{polylog}(m)$ space, where $m$ is the size of the data, e.g., the length of the data stream. The problem can be solved in $O(\log m)$ space in the random access model. For deterministic algorithms, (Tarui, 2007) showed that even if the streaming algorithm is allowed $k$ passes over the data, $\Omega(m^{1/k})$ memory is necessary. (Gopalan & Radhakrishnan, 2009) answered the question of (Muthukrishnan, 2005) in the affirmative by giving a randomized algorithm that uses $O(\log^3(m))$ bits of space for the duplication detection problem when the stream length is larger than the universe size. Without such a condition there is a strong $\Omega(m)$ memory lower bound. (Jowhari et al., 2011) finally gave tight upper and lower bounds, showing that the space complexity of the duplication detection problem is $\Theta(\log^2(m))$ bits.

## C.2. Hardness of Duplication Detection Problem

In this section, we give our lower bound for the communication complexity of the duplication detection problem, which we recall is defined as follows: let $C$ be an input parameter for the number of collisions and $k$ be an input parameter for the number of players with mutual collisions. Let $\varepsilon \in (0, 1)$ be an accuracy parameter. Suppose there exist $\alpha$ players, each with $s$ samples. The goal is to identify whether there are fewer than $C$ coordinates or more than $(1 + \varepsilon) \cdot C$ coordinates shared among exactly $k$ players.

Our approach is to reduce the problem of duplication detection from the GapSet problem. We split the analysis into casework depending on whether $C = O\left(\frac{1}{\varepsilon^2}\right)$ or $C = \Omega\left(\frac{1}{\varepsilon^2}\right)$. However, we first require the following structural properties to argue distribution matching.

**Fact C.1** (Stirling's approximation). *For any integer $k > 0$, we have*

$$\sqrt{2\pi}k^{k+\frac{1}{2}}e^{-k} \leq k! \leq ek^{k+\frac{1}{2}}e^{-k}.$$

**Lemma C.2.** *Let $c > 0$ be a constant. Let $X$ be a random variable drawn from the binomial distribution on $t$ trials with probability $\frac{1}{4}$. Then we have*

$$\mathbf{Pr}\left[\left|X - \frac{t}{4}\right| < c \cdot \sqrt{t}\right] \leq c.$$

*Proof.* We assume that $t$ is divisible by $4$ and define the interval

$$I = \left\{i \in \mathbb{N} \mid \frac{t}{4} - c \cdot \sqrt{t} < i < \frac{t}{4} + c \cdot \sqrt{t}\right\}.$$

Note that if $t$ is not divisible by $4$, we can perform a similar analysis on $4\left\lceil\frac{t}{4}\right\rceil$.

Let $X$ be a random variable drawn from the binomial distribution on $t$ trials with probability $\frac{1}{4}$. We bound the probability $\mathbf{Pr}\left[X \in I\right] = \sum_{i \in I} \mathbf{Pr}\left[X = i\right]$. Observe that we have

$$\mathbf{Pr}\left[X = i\right] = \left(\frac{1}{4}\right)^i \left(\frac{3}{4}\right)^{t-i} \binom{t}{i}.$$

Hence,

$$\frac{\mathbf{Pr}\left[X = i\right]}{\mathbf{Pr}\left[X = i-1\right]} = \frac{1}{3}\frac{(i-1)!(t-i+1)!}{i!(t-i)!} = \frac{1}{3}\frac{t-i+1}{i},$$

so that $\frac{\mathbf{Pr}[X=i]}{\mathbf{Pr}[X=i-1]} < 1$ for $i > \frac{t}{4}$ and $\frac{\mathbf{Pr}[X=i]}{\mathbf{Pr}[X=i-1]} > 1$ for $i \leq \frac{t}{4}$. Therefore, $\mathbf{Pr}\left[X = i\right]$ is maximized at $i = \frac{t}{4}$.

We now upper bound $\binom{t}{t/4}$ using Stirling's approximation in Fact C.1 to handle the binomial coefficients. We have

$$\binom{t}{t/4} = \frac{t!}{(3t/4)!(t/4)!}$$

$$\leq \frac{et^{t+\frac{1}{2}}e^{-t}}{\left(\sqrt{2\pi}(3t/4)^{3t/4+1/2}e^{-3t/4}\right)\left(\sqrt{2\pi}(t/4)^{t/4+1/2}e^{-t/4}\right)}$$

$$= \frac{e}{2\pi\sqrt{t}\,(3/4)^{3t/4}\,(1/4)^{t/4}}.$$

Hence, we have

$$\mathbf{Pr}\left[X = \frac{t}{4}\right] \leq \left(\frac{1}{4}\right)^{t/4}\left(\frac{3}{4}\right)^{3t/4}\frac{e}{2\pi\sqrt{t}\,(3/4)^{3t/4}\,(1/4)^{t/4}}$$

$$= \frac{e}{2\pi\sqrt{t}} < \frac{1}{2\sqrt{t}}.$$

Therefore,

$$\mathbf{Pr}\left[X \in I\right] \leq \sum_{i \in I} \mathbf{Pr}\left[X = i\right] < \sum_{i \in I} \frac{1}{2\sqrt{t}} = \frac{|I|}{2\sqrt{t}} \leq \frac{2c\sqrt{t}}{2\sqrt{t}} = c.$$

$\square$

We first consider the case where $C < \frac{4}{\varepsilon^2}$.

**Lemma C.3.** *Let $C$ be an input parameter for the number of collisions and $k$ be an input parameter for the number of players with mutual collisions. Let $\varepsilon \in (0, 1)$ be an accuracy parameter such that $C < \frac{4}{\varepsilon^2}$. Suppose there exist $\alpha$ players, each with $s$ samples from some universe of size $N = \Omega(s)$. Then any protocol $\Pi$ that with probability at least $0.99$, identifies whether there are fewer than $(1 - \varepsilon) \cdot C$ coordinates or more than $(1 + \varepsilon) \cdot C$ coordinates shared among exactly $k$ players requires $\Omega(\alpha s)$ communication.*

*Proof.* We first consider the setting where $C < \frac{4}{\varepsilon^2}$. Let $t = 4C$ and $n = \Omega\left(\frac{\alpha s}{C}\right)$. Let $\{\mathbf{u}^{(i)}\}_{i \in [\alpha]}$ be an instance of $\mathsf{GapSet}_{t,\alpha,n,k}$. Recall that $\{\mathbf{u}^{(i)}\}_{i \in [\alpha]}$ is generated from $\mathbf{x}, \mathbf{y} \in \{0, 1\}^t$ drawn from $\mathsf{GapAnd}_t$. For each $j \in [t]$, let $D_j$ denote the indicator random variable for whether $x_j = y_j = 1$, so that $D_j = 1$ if $x_j = y_j = 1$ and $D_j = 0$ otherwise. Note that $D_j$ is a Bernoulli random variable with parameter $\frac{1}{4}$. Let $D = \sum_{j \in [t]} D_j$, so that $D$ is a binomial random variable with $t$ trials and parameter $\frac{1}{4}$.

Observe that $\frac{1}{16}\sqrt{t} \geq \varepsilon\frac{t}{4}$, so that a $(1 + \varepsilon)$-approximation algorithm to the number of $k$-wise collisions will also determine the number of coordinates $j \in [t]$ such that $D_j = 1$. We have that $\Pr\left[\left|D - \frac{t}{4}\right| \geq \frac{1}{16}\sqrt{t}\right] \geq 0.2$. Thus we have that with probability at least $\frac{3}{4}$, $\Pi$ will be able to solve $\mathsf{GapSet}_{t,\alpha,n,k}$. Hence by Theorem B.13, $\Pi$ must use $\Omega(nt) = \Omega(\alpha s)$ communication. $\square$

We next consider the case where $C \geq \frac{4}{\varepsilon^2}$. The proof follows similarly to Lemma C.3 but also uses a padding argument to account for the number of collisions.

**Lemma C.4.** *Let $C$ be an input parameter for the number of collisions and $k$ be an input parameter for the number of players with mutual collisions. Let $\varepsilon \in (0, 1)$ be an accuracy parameter such that $C \geq \frac{4}{\varepsilon^2}$. Suppose there exist $\alpha$ players, each with $s$ samples from some universe of size $N = \Omega(s)$. Then any protocol $\Pi$ that with probability at least $0.99$, identifies whether there are fewer than $(1 - \varepsilon) \cdot C$ coordinates or more than $(1 + \varepsilon) \cdot C$ coordinates shared among exactly $k$ players requires $\Omega\left(\frac{\alpha s}{C\varepsilon^2}\right)$ communication.*

*Proof.* Let $t = \frac{1}{\varepsilon^2}$ and $n = \Omega\left(\frac{\alpha s}{C}\right)$. Let $\{\mathbf{v}^{(i)}\}_{i \in [\alpha]}$ be an instance of $\mathsf{GapSet}_{t,\alpha,n,k}$. We then set $\mathbf{u}^{(i)}$ to be $\frac{C}{t}$ copies of $\mathbf{v}^{(i)}$ for each $i \in [\alpha]$, so that $\mathbf{u}^{(i)} = \mathbf{v}^{(i)} \circ \mathbf{v}^{(i)} \circ \ldots \circ \mathbf{v}^{(i)}$. For each $j \in [t]$, we define $D_j$ to be the indicator random variable for whether $x_j = y_j = 1$, so that $D_j = 1$ if $x_j = y_j = 1$ and $D_j = 0$ otherwise. Since $\{\mathbf{v}^{(i)}\}_{i \in [\alpha]}$ is generated from $\mathbf{x}, \mathbf{y} \in \{0, 1\}^t$ drawn from $\mathsf{GapAnd}_t$, then $D_j$ is a Bernoulli random variable with parameter $\frac{1}{4}$. Hence for $D = \sum_{j \in [t]} D_j$, we have that $D$ is a binomial random variable with $t$ trials and parameter $\frac{1}{4}$.

Observe that $\frac{1}{16}\sqrt{t} \geq \varepsilon\frac{t}{4}$, so that a $(1 + \varepsilon)$-approximation algorithm to the number of $k$-wise collisions will also determine the number of coordinates $j \in [t]$ such that $D_j = 1$. By Lemma C.2, we have that $\mathbf{Pr}\left[\left|D - \frac{t}{4}\right| \geq \frac{1}{16}\sqrt{t}\right] \geq 0.2$. Thus we have that with probability at least $\frac{3}{4}$, $\Pi$ will be able to solve $\mathsf{GapSet}_{t,\alpha,n,k}$. Hence by Theorem B.13, $\Pi$ must use $\Omega(nt) = \Omega\left(\frac{\alpha s}{C\varepsilon^2}\right)$ communication. $\square$

Putting Lemma C.3 and Lemma C.4 together, we have:

**Theorem C.5.** *Let $C$ be an input parameter for the number of collisions, $k$ be an input parameter for the number of players with mutual collisions, and $\varepsilon \in (0, 1)$ be an accuracy parameter. Suppose there exist $\alpha$ players, each with $s$ samples from some universe of size $N = \Omega(s)$. Then any protocol $\Pi$ that with probability at least $0.99$, identifies whether there are fewer than $(1 - \varepsilon) \cdot C$ coordinates or more than $(1 + \varepsilon) \cdot C$ coordinates shared among exactly $k$ players requires $\Omega(\alpha s)$ communication for $C < \frac{4}{\varepsilon^2}$ and $\Omega\left(\frac{\alpha s}{C\varepsilon^2}\right)$ communication for $C \geq \frac{4}{\varepsilon^2}$.*

Theorem 1.5 then follows from setting $k = 2$.

---

(1) For each player $i \in [\alpha]$, let $S_i$ be the set of items given to player $i$.

(2) For each $j \in [N]$, sample $j$ into a set $U$ with probability $p = \Theta\left(\frac{1}{C\varepsilon^2}\right)$. This is done by all players using public randomness.

(3) For each $i \in [\alpha]$, let $T_i = S_i \cap U$.

(4) Let $\mathbf{v}$ be a bit vector of size $\frac{\xi\alpha s}{C\varepsilon^2}$, for a sufficiently large constant $\xi > 0$.

(5) While there are multiple items in $\cup T_i$ that hash to the same position of $\mathbf{v}$:

    (a) All players hash $T_i$ into $\mathbf{v}$ and communicate the non-zero entries of their hash.

    (b) For any $x$ that does not map to a position of $\mathbf{v}$ communicated by multiple players, remove $x$ from the remaining items, i.e., $T_i = T_i \setminus \{x\}$ for all $i \in [\alpha]$.

(6) Let $D'$ be the number of positions that are communicated by multiple players.

(7) Output $\frac{D'}{p}$ for the estimated number of collisions.

---

Figure 5: Distributed protocol for duplication estimation

## C.3. Upper Bounds for Duplication Detection

In this section, we provide a short sketch of a distributed protocol for the duplication detection problem. The protocol uses standard techniques and is summarized in Figure 5. Given $C > 0$ and an accuracy parameter $\varepsilon \in (0, 1)$, the $\alpha$ parties must determine whether there exist at least $C \cdot (1 + \varepsilon)$ duplicates or at most $C \cdot (1 - \varepsilon)$ duplicates.

The protocol proceeds as follows. The parties first sample each item from the universe at a rate $p = \Theta\left(\frac{1}{C\varepsilon^2}\right)$. That is, instead of considering the universe $[N]$, they consider a universe $U$ where for each $i \in [N]$, we have $i \in U$ with probability $p$. Then each player only considers the subset of their items that are contained in $U$.

By standard calculations on the expectation and variance, it can be shown that if $D'$ is the number of duplicates across the subsampled universe $U$, then $\frac{1}{D'}$ is an additive $O(\varepsilon) \cdot C$ approximation to the actual number $D$ of duplicates. That is, with probability 0.99, we have $\left|\frac{D'}{p} - D\right| \leq O(\varepsilon) \cdot C$. It thus remains to compute the number of duplicates in the universe $U$.

By Markov's inequality, with probability at least 0.99, the total number of items in the universe $U$ is at most $\frac{\gamma\alpha s}{C\varepsilon^2}$ for some sufficiently large constant $\gamma$. Let $\mathcal{E}$ be the event that the total number of items in the universe $U$ is at most $\frac{\gamma\alpha s}{C\varepsilon^2}$, so that $\mathbf{Pr}[\mathcal{E}] \geq 0.99$. The players first hash their items into a bit vector $\mathbf{v}$ of size $\frac{\xi\alpha s}{C\varepsilon^2}$, for a sufficiently large constant $\xi > 0$. We call an item $i \in [N]$ isolated if it is hashed into a coordinate of $\mathbf{v}$ that no other item is hashed to. Note that conditioned on $\mathcal{E}$, the probability that each item is isolated is at least 0.999, for sufficiently large $\xi$.

The players then succinctly communicate the hashes of all their items as follows. For each $i \in [\alpha]$, let $N_i$ be the number of samples that player $i$ has that is contained in the universe $U$. To communicate their items, it suffices to use total communication

$$\sum_{i=1}^{\alpha} \log \binom{\gamma\alpha s/C\varepsilon^2}{N_i} = \log \prod_{i=1}^{\alpha} \binom{\gamma\alpha s/C\varepsilon^2}{N_i} \leq \log \prod_{i=1}^{\alpha} \binom{\gamma\alpha s/C\varepsilon^2}{\gamma s/C\varepsilon^2},$$

where the last inequality holds due to the constraint that conditioned on $\mathcal{E}$, we have

$$N_1 + \ldots + N_\alpha \leq \frac{\gamma\alpha s}{C\varepsilon^2}.$$

Thus the total communication is at most

$$\sum_{i=1}^{\alpha} \log \binom{\gamma\alpha s/C\varepsilon^2}{\gamma s/C\varepsilon^2} = \alpha \log \alpha \cdot O\left(\frac{\gamma s}{C\varepsilon^2}\right) = O\left(\frac{\gamma\alpha s \log \alpha}{C\varepsilon^2}\right).$$

Note that the central server will immediately observe that any isolated item cannot be duplicated. On the other hand, there could be multiple items that are not duplicated, yet are sent to the same coordinate in the bit vector $\mathbf{v}$. By Markov's

inequality, we have that with probability at least 0.99, half of the items that are not duplicates are isolated. It then suffices to recurse.

That is, in each iteration $\ell$, suppose we have $\frac{\gamma_\ell \alpha s}{C\varepsilon^2}$ remaining items that are not duplicated. Then running the above protocol, the total communication in round $\ell$ is $O\left(\frac{\gamma_\ell \alpha s \log \alpha}{C\varepsilon^2}\right)$. We have that $\mathbb{E}[\gamma_\ell] \leq \frac{1}{2}\mathbb{E}[\gamma_{\ell-1}]$. Thus, in expectation, the total communication is a geometric series that sums to $O\left(\frac{\gamma \alpha s \log \alpha}{C\varepsilon^2}\right)$. Then again by Markov's inequality and the fact that $\gamma$ is a constant, we have that the total communication is $O\left(\frac{\alpha s \log \alpha}{C\varepsilon^2}\right)$.

**Theorem C.6.** *Given $C > 0$ and an accuracy parameter $\varepsilon \in (0,1)$, there exists a distributed protocol for $\alpha$ parties that determine whether there exist at least $C \cdot (1+\varepsilon)$ duplicates or at most $C \cdot (1-\varepsilon)$ duplicates with probability at least $\frac{2}{3}$ and uses $O\left(\frac{\alpha s \log \alpha}{C\varepsilon^2}\right)$ communication.*

*Proof.* Let $D$ be the number of duplicates across the $\alpha$ parties and $D'$ be the number of duplicates in the subsampled universe. Let $\mathcal{D}$ be the set of duplicates among the $\alpha$ parties, so that $|\mathcal{D}| = D$. Let $p$ be the probability that each coordinate in the universe is sampled. Then we have

$$\mathbb{E}[D'] = \sum_{i \in \mathcal{D}} p = Dp,$$

so that $\mathbb{E}\left[\frac{D'}{p}\right] = D$. Moreover, we have

$$\mathbb{V}\left[\frac{D'}{p}\right] \leq \frac{1}{p^2}\sum_{i \in \mathcal{D}} p = \frac{D}{p} = O\left(C\varepsilon^2 D\right),$$

so that by Chebyshev's inequality, we have

$$\mathbf{Pr}\left[\left|\frac{D'}{p} - D\right| \leq \varepsilon \cdot O\left(\sqrt{CD}\right)\right] \geq 0.99.$$

Now we note that if $D < \frac{C}{1000}$ or $D > 1000C$, then a simple 100-approximation to $D$ suffices to distinguish whether $D > C \cdot (1+\varepsilon)$ or $D < C \cdot (1-\varepsilon)$. Thus we assume $\frac{C}{1000} \leq D \leq 1000C$, so that

$$\mathbf{Pr}\left[\left|\frac{D'}{p} - D\right| \leq O\left(\varepsilon\right) \cdot C\right] \geq 0.99.$$

It thus remains to compute the number of duplicates in the universe $U$. To that end, the players first hash their items into a bit vector $\mathbf{v}$ of size $\frac{\xi \alpha s}{C\varepsilon^2}$, for a sufficiently large constant $\xi > 0$.

We call an item $i \in [N]$ isolated if it is hashed into a coordinate of $\mathbf{v}$ that no other item is hashed to. Note that conditioned on $\mathcal{E}$, the probability that each item is isolated is at least 0.999, for sufficiently large $\xi$. By Markov's inequality, with probability at least 0.99, the total number of items in the universe $U$ is at most $\frac{\gamma \alpha s}{C\varepsilon^2}$ for some sufficiently large constant $\gamma$. Let $\mathcal{E}$ be the event that the total number of items in the universe $U$ is at most $\frac{\gamma \alpha s}{C\varepsilon^2}$, so that $\mathbf{Pr}[\mathcal{E}] \geq 0.99$.

The players then succinctly communicate the hashes of all their items as follows. For each $i \in [\alpha]$, let $N_i$ be the number of samples that player $i$ has that is contained in the universe $U$. To communicate their items, it suffices to use total communication

$$\sum_{i=1}^{\alpha} \log \binom{\gamma \alpha s/C\varepsilon^2}{N_i} = \log \prod_{i=1}^{\alpha} \binom{\gamma \alpha s/C\varepsilon^2}{N_i} \leq \log \prod_{i=1}^{\alpha} \binom{\gamma \alpha s/C\varepsilon^2}{\gamma s/C\varepsilon^2},$$

where the last inequality holds due to the constraint that conditioned on $\mathcal{E}$, we have

$$N_1 + \ldots + N_\alpha \leq \frac{\gamma \alpha s}{C\varepsilon^2}.$$

Thus the total communication is at most

$$\sum_{i=1}^{\alpha} \log \binom{\gamma \alpha s/C\varepsilon^2}{\gamma s/C\varepsilon^2} = \alpha \log \alpha \cdot O\left(\frac{\gamma s}{C\varepsilon^2}\right) = O\left(\frac{\gamma \alpha s \log \alpha}{C\varepsilon^2}\right).$$

Note that the central server will immediately observe that any isolated item cannot be duplicated. On the other hand, there could be multiple items that are not duplicated, yet are sent to the same coordinate in the bit vector $\mathbf{v}$. By Markov's inequality, we have that with probability at least $0.99$, half of the items that are not duplicates are isolated.

The protocol is then performed recursively. Specifically, in each iteration $\ell$, suppose there remain $\frac{\gamma_\ell \alpha s}{C \varepsilon^2}$ items that are not duplicated. Then running the above protocol, the total communication in iteration $\ell$ is at most $\frac{\tau \gamma_\ell \alpha s \log \alpha}{C \varepsilon^2}$ for some constant $\tau > 0$. Call an iteration *successful* if $\gamma_\ell \le \frac{1}{2} \gamma_{\ell-1}$, so that the above argument implies that each iteration is successful with probability at least $0.99$. Thus we have

$$\mathbb{E}\left[\gamma_\ell\right] \le \frac{99}{100} \frac{1}{2} \gamma_{\ell-1} + \frac{1}{100} \gamma_{\ell-1} \le \frac{2}{3} \gamma_{\ell-1}.$$

Then the expected communication $\Lambda_\ell$ of iteration $\ell + 1$ is at most

$$\mathbb{E}\left[\Lambda_\ell\right] \le \frac{\tau \gamma_\ell \alpha s \log \alpha}{C \varepsilon^2} \le \left(\frac{2}{3}\right)^\ell \frac{\tau \gamma \alpha s \log \alpha}{C \varepsilon^2}.$$

Thus, in expectation, the total communication $\Lambda$ is a geometric series that sums to $O\left(\frac{\gamma \alpha s \log \alpha}{C \varepsilon^2}\right)$:

$$\mathbb{E}\left[\Lambda\right] = \sum_\ell \mathbb{E}\left[\Lambda_\ell\right] \le O\left(\frac{\gamma \alpha s \log \alpha}{C \varepsilon^2}\right).$$

Then again by Markov's inequality and the fact that $\gamma$ is a constant, we have that the total communication is $O\left(\frac{\alpha s \log \alpha}{C \varepsilon^2}\right)$.  □

# D. Parameterized Streaming for Distinct Elements

In this section, we consider distinct elements estimation in the streaming model. Namely, there exists an underlying vector $x \in \mathbb{R}^n$ and each update in a stream of length $m = \mathrm{poly}(n)$ can increase or decrease a coordinate of $x$. The goal is to estimate $\|x\|_0$ within a multiplicative factor of $(1 + \varepsilon)$ at the end of the stream using space polylogarithmic in $n$. Moreover, it is known that $\omega\left(\frac{1}{\varepsilon^2}\right)$ bits of space is generally needed to solve this problem. We now show that if the number of coordinates with frequency more than one is small, this lower bound need not hold.

We first require the following guarantees of the well-known COUNTSKETCH data structure from streaming.

**Theorem D.1.** *(Charikar et al., 2002) Let $x \in \mathbb{R}^n$ and let $y$ with the vector $x$ with the $b$ coordinates largest in magnitude set to zero. Then with high probability, for each $i \in [n]$, COUNTSKETCH outputs an estimate $\widehat{x}_i$ such that*

$$|\widehat{x}_i - x_i| \le \frac{1}{\sqrt{b}} \|y\|_2.$$

*Moreover, COUNTSKETCH uses $O\left(b \log^2 n\right)$ bits of space.*

## D.1. Robust Statistics

In this section, we present a streaming algorithm for distinct element estimation based on robust statistics. We first recall the following statement from robust mean estimation.

**Theorem D.2.** *(Prasad et al., 2019) Let $P$ be any $2k$-moment bounded distribution over $\mathbb{R}$ with mean $\mu$ and variance bounded by $\sigma^2$. Let $Q$ be an arbitrary distribution and the mixture $P_\varepsilon = (1 - \varepsilon)P + \varepsilon Q$. Given $n$ samples from $P_\varepsilon$, there exists an algorithm ROBUSTMEANEST that returns an estimate $\widehat{\mu}$ such that with probability at least $1 - \delta$,*

$$|\widehat{\mu} - \mu| \le O\left(\sigma\right)\left(\max\left(\varepsilon, \frac{\log(1/\delta)}{n}\right)^{1 - \frac{1}{2k}} + \left(\frac{\log n}{n}\right)^{1 - \frac{1}{2k}} + \sqrt{\frac{\log 1/\delta}{n}}\right).$$

We next present the algorithm in Algorithm 3.

We show that sampling with probability $p$ so that there are $\Theta\left(\frac{1}{\varepsilon^2}\right)$ items in $S$ implies that $\frac{1}{p} \cdot |S|$ is roughly a $(1 + O\left(\varepsilon\right))$-approximation to the total number of distinct elements. The statement is well-known; we include the proof for completeness.

---

**Algorithm 3** Parameterized streaming algoritihm for distinct element estimation using robust statistics

---

**Input:** Accuracy parameter $\varepsilon \in (0, 1)$, number $C$ of coordinates that are greater than 1
**Output:** $(1 + \varepsilon)$-approximation to the number of distinct elements
 1: Let $X \in \left[ \frac{F_0}{100}, F_0 \right]$
 2: Let $S \subset [n]$ be formed by sampling each item of $[n]$ with probability $p = \min\left(1, \frac{1}{100\varepsilon^2 X}\right)$
 3: $B \leftarrow O\left(\frac{C}{\varepsilon}\right)$
 4: **for** $b \in [B]$ **do**
 5:      Let $S_b$ sample each item of $S$ with probability $\frac{1}{B}$
 6:      Let $f_b = F_1(S_b)$ be the total number of updates to items in $S_b$
 7: **end for**
 8: $Z \leftarrow B \cdot \text{ROBUSTMEANEST}(f_1, \ldots, f_b)$
 9: **Return** $\frac{1}{p} \cdot Z$

---

**Lemma D.3.** *With probability at least* $0.99$*, we have*

$$\left(1 - \frac{\varepsilon}{4}\right) \cdot F_0 \leq \frac{1}{p} \cdot |S| \leq \left(1 + \frac{\varepsilon}{4}\right) \cdot F_0.$$

*Proof.* Let $N$ be the set of distinct elements in the stream. Then we have

$$\mathbb{E}\left[|S|\right] = \frac{1}{p} \sum_{i \in N} p = |N|,$$

and

$$\mathbb{E}\left[|S|^2\right] \leq \frac{1}{p^2} \sum_{i \in N} p \leq \frac{|N|}{p} \leq \frac{\varepsilon^2}{10^6} \cdot |N|^2,$$

for $p \geq \frac{10^6 |N|}{\varepsilon^2}$. Therefore, by Chebyshev's inequality, we have with probability at least $0.99$,

$$\left| \frac{1}{p} \cdot |S| - |N| \right| \leq \frac{\varepsilon}{4} \cdot F_0.$$

$\square$

We now justify the correctness of Algorithm 3.

**Lemma D.4.** *If* $F_0(S) \geq \frac{1}{\varepsilon^2}$ *and* $C \leq \frac{1}{4\varepsilon}$*, then Algorithm 3 provides a* $(1 + \varepsilon)$*-approximation to the number of distinct elements in the stream with probability at least* $0.98$.

*Proof.* Let $f$ be the frequency vector defined over the stream and let $Z$ be defined as in Algorithm 3. Let $N_1$ be the set of items with frequency one in $S$ and $N_{>1}$ be the set of items with frequency larger than 1 in $S$. For any fixed $b \in [B]$, let $S_b(N_1)$ denote the subset of $N_1$ sampled into $S_b$ and similarly, let $S_b(N_{>1})$ denote the subset of $N_{>1}$ sampled into $S_b$. The probability that $|S_b(N_{>1})| = 0$ is $\left(1 - \frac{1}{B}\right)^C \leq \frac{\varepsilon}{4}$ for sufficiently large $B = O\left(\frac{C}{\varepsilon}\right)$. Moreover, the distribution of $|S_b(N_1)|$ is a binomial random variable with $N_1$ trials and $\frac{1}{B}$ success rate. Hence, $\mathbb{E}\left[|S_b(N_1)|\right] = \frac{1}{B} \cdot (|N_1|)$, $\mathbb{V}\left[|S_b(N_1)|\right] \leq \frac{1}{B} \cdot |N_1|$, and all moments of $|S_b(N_1)|$ are finite. Therefore, by the guarantees of ROBUSTMEANEST in Theorem D.2, we have that with high probability,

$$|Z - |N_1|| \leq B\sqrt{\frac{|N_1|}{B}} \cdot \frac{\varepsilon}{4} \leq \sqrt{\frac{C}{\varepsilon} \cdot F_0(S)} \cdot \frac{\varepsilon}{4} \leq \frac{\varepsilon}{4} \cdot F_0(S).$$

Since $F_0(S) \geq \frac{1}{\varepsilon^2}$ and $C \leq \frac{1}{4\varepsilon}$, then $|N_1|$ is a $\left(1 + \frac{\varepsilon}{4}\right)$-approximation to $F_0(S)$. Thus,

$$|Z - F_0(S)| \leq \frac{\varepsilon}{2} \cdot F_0(S).$$

Finally by Lemma D.3, we have with probability at least $0.99$,

$$\left(1 - \frac{\varepsilon}{4}\right) \cdot F_0 \leq \frac{1}{p} \cdot F_0(S) \leq \left(1 + \frac{\varepsilon}{4}\right) \cdot F_0,$$

so that with probability at least $0.98$,

$$\left| \frac{1}{p} \cdot Z - \|f\|_0 \right| \leq \varepsilon \cdot \|f\|_0.$$

$\square$

Next, we analyze the space complexity of Algorithm 3.

**Lemma D.5.** *For a stream with length polynomially bounded in $n$, Algorithm 3 uses $O\left(\frac{C}{\varepsilon} \log n\right)$ bits of space.*

*Proof.* Note that Algorithm 3 maintains $B = O\left(\frac{C}{\varepsilon}\right)$ buckets, each represented by a counter using $O(\log n)$ bits of space for a stream with length polynomially bounded in $n$. $\square$

Putting together the correctness of approximation and the space bounds, we have the following:

**Theorem D.6.** *Given an accuracy parameter $\varepsilon \in (0, 1)$, a parameter $C \leq \frac{1}{4\varepsilon}$ for the number of coordinates with frequency more than $1$, and a number of distinct elements that is at least $\Omega\left(\frac{1}{\varepsilon^2}\right)$, there exists a one-pass streaming algorithm that uses $O\left(\frac{C}{\varepsilon} \log n\right)$ bits of space and provides a $(1 + \varepsilon)$-approximation to the number of distinct elements in the stream with probability at least $0.98$.*

### D.2. Subsampling

In this section, we present a two-pass streaming algorithm based on subsampling. We give an algorithm for when the number of coordinates with frequency greater than one is relatively "large" in Algorithm 4 and for the case where the number of coordinates with frequency greater than one is "small" in Algorithm 5.

---

**Algorithm 4** Parameterized distinct element estimation over two-pass streams

---

**Input:** Accuracy parameter $\varepsilon \in (0, 1)$, number $C$ of coordinates that are greater than $1$
**Output:** $(1 + \varepsilon)$-approximation to the number of distinct elements, given two passes over the data
1: $L \leftarrow O\left(\log \frac{1}{\varepsilon}\right)$, $B \leftarrow C \cdot \text{polylog}\left(\frac{n}{\varepsilon}\right)$, $T \leftarrow \frac{100}{\varepsilon^2} \log^2 \frac{n}{\varepsilon}$
2: **for** $\ell \in [L]$ **do**
3:     Form $S_\ell$ by sampling each item of $[n]$ with probability $\frac{1}{2^{2\ell-2}}$
4:     Run $O(\log n)$ instances of COUNTSKETCH on $S_\ell$ with $B$ buckets         ▷First pass
5: **end for**
6: **for** each heavy-hitter $i \in [n]$ reported by COUNTSKETCH on any $S_\ell$ **do**
7:     Track $f_i$ exactly         ▷Second pass
8: **end for**
9: **for** $\ell \in [L]$ **do**
10:     $\widehat{M_\ell} = 0$
11:     **for** $j \in S_\ell$ **do**
12:         **if** $f_j \geq T$ **then**
13:             $\widehat{M_1} \leftarrow \widehat{M_1} + (f_j - 1)$
14:         **else if** $f_j \in \left[\frac{T}{2^\ell}, \frac{T}{2^{\ell-1}}\right)$ **then**
15:             $\beta \leftarrow \max\left(0, \ell - \log\left(10\left(\frac{\sqrt{C}}{\varepsilon}\right)\log\frac{n}{\varepsilon}\right)\right)$
16:             $\widehat{M_\ell} \leftarrow \widehat{M_\ell} + 2^{2\beta} \cdot (f_j - 1)$
17:         **end if**
18:     **end for**
19: **end for**
20: **Return** $F_1(S_1) - \sum_{\ell \in [L]} \widehat{M_\ell}$

---

**Lemma D.7.** *Let $Z$ be the output of Algorithm 4. Then with probability at least $\frac{2}{3}$, we have that*

$$|Z - F_0(S)| \le \varepsilon F_0(S).$$

*Proof.* Let $S$ be the data stream. For each $i \in [n]$, let $m_i = \max(0, f_i - 1)$ so that $t_i$ is the excess mass of $f_i$. Let $M = \sum_{i \in [n]} m_i$ so that $F_0(S) = F_1(S) - M$. Thus to achieve a $(1 + \varepsilon)$-approximation to $F_0(S)$, it suffices to obtain an additive $\varepsilon \cdot F_0$ approximation to $M$.

Let level set $\Gamma_1 = \left\{ i \in [n] : f_i \ge \frac{T}{2} \right\}$ consist of the coordinates $i \in [n]$ with frequency at least $\frac{T}{2}$. Similarly, for $\ell > 1$, let level set $\Gamma_\ell = \left\{ i \in [n] : \left[ \frac{T}{2^\ell}, \frac{T}{2^{\ell-1}} \right) \right\}$ consist of the coordinates $i \in [n]$ with frequency in the interval $\left[ \frac{T}{2^\ell}, \frac{T}{2^{\ell-1}} \right)$. Let $M_\ell = \sum_{i \in \Gamma_\ell} f_i$ be the sum of the contributions of the items in level set $\Gamma_\ell$. Finally, let $\Gamma$ be the set of all coordinates with value greater than 1.

For each $i \in M_\ell$, we have that $i$ is sampled with probability $p_\ell = \frac{1}{2^{\beta_\ell}}$, where

$$\beta_\ell = \max\left( 0, \ell - \log\left( 10 \left( \frac{\sqrt{C}}{\varepsilon} \right) \log \frac{n}{\varepsilon} \right) \right).$$

Hence, we consider casework on whether $\ell \le \log\left( 10 \left( \frac{\sqrt{C}}{\varepsilon} \right) \log \frac{n}{\varepsilon} \right)$ or whether $\ell > \log\left( 10 \left( \frac{\sqrt{C}}{\varepsilon} \right) \log \frac{n}{\varepsilon} \right)$.

Suppose $\ell \le \log\left( 10 \left( \frac{\sqrt{C}}{\varepsilon} \right) \log \frac{n}{\varepsilon} \right)$, so that $p_\ell = 1$. Then $i \in S_\ell$ for any $i \in \Gamma_\ell$. Let $\mathcal{G}_r$ denote the probability that $i$ is not hashed by the $r$-th instance of COUNTSKETCH to a bucket containing any of the other items in $\Gamma$, so that we have $\mathbf{Pr}\left[ \mathcal{E}_r \right] \ge \frac{2}{3}$ since $|\Gamma| \le C$ and we use $B = C \cdot \text{polylog}\left( \frac{n}{\varepsilon} \right)$ buckets in each instance of COUNTSKETCH. Then let $\mathcal{E}$ denote the probability that for all items $i \in S_\ell \cap \Gamma_\ell$, there exist $O(\log n)$ instances of COUNTSKETCH such that $i$ is not hashed to a bucket containing any of the other items in $\Gamma_\ell \cap S_\ell$, so that we have $\mathbf{Pr}\left[ \mathcal{E} \right] = \mathbf{Pr}\left[ \mathcal{G}_1 \vee \mathcal{G}_2 \vee \ldots \right] \ge 1 - \frac{1}{\text{poly}(n)}$. Conditioning on $\mathcal{G}_r$, the variance for the estimation of $f_i$ by the $r$-th instance COUNTSKETCH is at most $\frac{1}{B} \cdot \frac{100}{\varepsilon^2}$. Since $f_i \ge \frac{T}{2^{\ell-1}} \ge \frac{1}{C\varepsilon^2} \log \frac{n}{\varepsilon}$ then COUNTSKETCH reports $f_i$ as a heavy-hitter with probability $\frac{2}{3}$. Therefore, we have that with high probability, $i$ is reported as a heavy-hitter by some instance of COUNTSKETCH. Hence by a union bound over all $i \in S_\ell$, we have that with high probability, $\widehat{M_\ell} = M_\ell$.

Next, we suppose $\ell > \log\left( 10 \left( \frac{\sqrt{C}}{\varepsilon} \right) \log \frac{n}{\varepsilon} \right)$, so that $p_\ell = \frac{1}{2^{\beta_\ell}}$, where

$$\beta_\ell = \max\left( 0, \ell - \log\left( 10 \left( \frac{\sqrt{C}}{\varepsilon} \right) \log \frac{n}{\varepsilon} \right) \right).$$

Then for any $i \in \Gamma_\ell$, we have $i \in S_\ell$ with probability $p_\ell$. Let $\mathcal{E}_1$ denote the event that $F_0(S_\ell) \le (10 \log n) \cdot p_\ell \cdot F_0(S)$ so that $\mathbf{Pr}\left[ \mathcal{E}_1 \right] \ge 1 - \frac{1}{10n^2}$. Again, let $\mathcal{G}_r$ denote the event that $i \in S_\ell$ is not hashed by the $r$-th instance of COUNTSKETCH to a bucket containing any of the other items in $\Gamma$, so that we have $\mathbf{Pr}\left[ \mathcal{E}_r \right] \ge \frac{2}{3}$ since $|\Gamma| \le C$ and we use $B = C \cdot \text{polylog}\left( \frac{n}{\varepsilon} \right)$ buckets in each instance of COUNTSKETCH. Moreover, let $\mathcal{E}_2$ denote the probability that for all items $i \in S_\ell \cap \Gamma_\ell$, there exist $O(\log n)$ instances of COUNTSKETCH such that $i$ is not hashed to a bucket containing any of the other items in $\Gamma_\ell \cap S_\ell$. Then we have $\mathbf{Pr}\left[ \mathcal{E}_2 \right] = \mathbf{Pr}\left[ \mathcal{G}_1 \vee \mathcal{G}_2 \vee \ldots \right] \ge 1 - \frac{1}{\text{poly}(n)}$. Conditioning on $\mathcal{E}_1$ and $\mathcal{E}_r$, the variance for the estimation of $f_i$ by the $r$-th instance COUNTSKETCH is at most $\frac{1}{B} \cdot \frac{100}{\varepsilon^2} \cdot (10 \log n) \cdot p_\ell$. Since $f_i \ge \frac{T}{2^{\ell-1}} \ge \frac{10}{\varepsilon} \log \frac{n}{\varepsilon}$ then COUNTSKETCH reports $f_i$ as a heavy-hitter with probability at least $\frac{2}{3}$. Therefore, we have

$$\mathbb{E}\left[ \widehat{M_\ell} \right] = \frac{1}{p_\ell} \sum_{j \in \Gamma_\ell} p_\ell \cdot f_j = M_\ell.$$

Furthermore,

$$\mathbb{E}\left[ (\widehat{M_\ell})^2 \right] \le \frac{1}{p_\ell^2} \sum_{j \in \Gamma_\ell} p_\ell \cdot f_j^2 \le C \cdot \frac{T^2}{2^{2\ell}} \cdot \frac{\varepsilon^2 \cdot 2^{2\ell}}{\gamma C T^2 \log^2\left( \frac{n}{\varepsilon} \right)} \le \frac{\varepsilon^2}{\gamma \log^2\left( \frac{n}{\varepsilon} \right)} \cdot F_0^2(S),$$

for some large constant $\gamma > 1$. Thus by Chebyshev's inequality, we have that with probability at least $1 - \frac{1}{100 \log \frac{n}{\varepsilon}}$,

$$|M_\ell - \widehat{M_\ell}| \le \frac{\varepsilon}{L} \cdot F_0.$$

The result then follows from union bounding over all $L$ level sets $\Gamma_1, \ldots, \Gamma_L$. $\qquad\square$

To complete the guarantees of Algorithm 4, it remains to analyze the space complexity.

**Theorem D.8.** *Given a stream $S$, an accuracy parameter $\varepsilon \in (0, 1)$, a parameter $C \geq \frac{1}{\varepsilon} \cdot F_0(S)$ for the number of coordinates with frequency more than $1$, and a number of distinct elements that is at least $F_0(S) = \Omega\left(\frac{1}{\varepsilon^2}\right)$, there exists a two-pass streaming algorithm that uses $C \cdot \text{polylog}\left(\frac{n}{\varepsilon}\right)$ bits of space and provides a $(1 + \varepsilon)$-approximation to the number of distinct elements in the stream with probability at least $0.98$.*

*Proof.* The proof of correctness follows from Lemma D.7. The space complexity follows from the fact that we maintain $B$ buckets in each of the $O(\log n)$ instances of COUNTSKETCH, for $B = C \cdot \text{polylog}\left(\frac{1}{\varepsilon}\right)$. □

We now show that for $C > \frac{1}{\varepsilon}$, any algorithm for $(1 + \varepsilon)$-approximation to distinct elements requires $\Omega(C)$ bits of space. Recall that in Gap-Hamming problem, Alice is given binary vector $X \in \{0, 1\}^n$ and Bob is given binary vector $Y \in \{0, 1\}^n$ and the goal is to determine whether the Hamming distance between $X$ and $Y$ is either at least $\frac{n}{2} + \sqrt{n}$ or at most $\frac{n}{2} - \sqrt{n}$.

**Theorem D.9.** *(Chakrabarti & Regev, 2012) Any communication protocol that solves the Gap-Hamming problem with probability at least $\frac{2}{3}$ requires $\Omega(n)$ bits of communication.*

**Theorem D.10.** *For any frequency vector that has the number $C = \Omega\left(\frac{1}{\varepsilon}\right)$ of coordinates with frequency more than $1$, any one-pass streaming algorithm for $(1 + \varepsilon)$-approximation of the number of distinct elements must use $\Omega\left(\min\left(\frac{1}{\varepsilon^2}, C\right)\right)$ bits of space.*

*Proof.* Consider an instance of Gap-Hamming that has $n = \Theta(C)$ coordinates, where $C = \Omega\left(\frac{1}{\varepsilon}\right)$. Note that for the purposes of the proof, it suffices to assume that $C = O\left(\frac{1}{\varepsilon^2}\right)$. Namely, let $X$ be the input vector to Alice and let $Y$ be the input vector to Bob. Then $Z := X + Y$ has $O(C)$ coordinates with frequency more than $1$. Moreover, any $(1 + \varepsilon)$-approximation to $F_0(Z)$ will distinguish whether the Hamming distance between $X$ and $Y$ is at least $\frac{n}{2} + \sqrt{n}$ or less than $\frac{n}{2} - \sqrt{n}$, since $n = O\left(\frac{1}{\varepsilon^2}\right)$. Therefore, such an algorithm can be used to solve Gap-Hamming on $\Theta(C)$ coordinates and by Theorem D.9 must use space $\Omega(C)$. □

---

**Algorithm 5** Parameterized distinct element estimation over two-pass streams

---

**Input:** Accuracy parameter $\varepsilon \in (0, 1)$, stream with small number $C$ of coordinates with frequency larger than $1$, i.e., $C < \frac{1}{\varepsilon}$
**Output:** $(1 + \varepsilon)$-approximation to the number of distinct elements, given two passes over the data
 1: $L \leftarrow O\left(\log\frac{1}{\varepsilon}\right)$, $B \leftarrow \frac{1}{\varepsilon} \cdot \text{polylog}\left(\frac{n}{\varepsilon}\right)$, $T \leftarrow \frac{100}{\varepsilon^2}\log^2\frac{n}{\varepsilon}$
 2: **for** $\ell \in [L]$ **do**
 3:      Form $S_\ell$ by sampling each item of $[n]$ with probability $\frac{1}{2^{\ell-1}}$
 4:      Run $O(\log n)$ instances of COUNTSKETCH on $S_\ell$ with $B$ buckets            ▷First pass
 5: **end for**
 6: **for** each heavy-hitter $i \in [n]$ reported by COUNTSKETCH on any $S_\ell$ **do**
 7:      Track $f_i$ exactly                                                  ▷Second pass
 8: **end for**
 9: **for** $\ell \in [L]$ **do**
10:      $\widehat{M_\ell} = 0$
11:      **for** $j \in S_\ell$ **do**
12:          **if** $f_j \geq T$ **then**
13:              $\widehat{M_1} \leftarrow \widehat{M_1} + (f_j - 1)$
14:          **else if** $f_j \in \left[\frac{T}{2^\ell}, \frac{T}{2^{\ell-1}}\right)$ **then**
15:              $\beta \leftarrow \max\left(0, \ell - \log\left(\frac{10}{\varepsilon}\log\frac{n}{\varepsilon}\right)\right)$
16:              $\widehat{M_\ell} \leftarrow \widehat{M_\ell} + 2^\beta \cdot (f_j - 1)$
17:          **end if**
18:      **end for**
19: **end for**
20: **Return** $F_1(S_1) - \sum_{\ell \in [L]} \widehat{M_\ell}$

---

We now justify the correctness of Algorithm 5.

**Lemma D.11.** *Let $Z$ be the output of Algorithm 5 and suppose the number $C$ of pairwise collisions is at most $\frac{1}{\varepsilon}$. Then with probability at least $\frac{2}{3}$, we have that*

$$|Z - F_0(S)| \leq \varepsilon F_0(S).$$

*Proof.* Let $S$ be the data stream and for each $i \in [n]$, let $m_i = \max(0, f_i - 1)$ so that $t_i$ is the excess mass of $f_i$. Let $M = \sum_{i \in [n]} m_i$ so that $F_0(S) = F_1(S) - M$. To achieve a $(1+\varepsilon)$-approximation to $F_0(S)$, it suffices to obtain an additive $\varepsilon \cdot F_0$ approximation to $M$.

Let level set $\Gamma_1 = \left\{ i \in [n] : f_i \geq \frac{T}{2} \right\}$ comprise the coordinates $i \in [n]$ with frequency at least $\frac{T}{2}$. Now for integer $\ell \in (1, L)$, we define level set $\Gamma_\ell = \left\{ i \in [n] : \left[ \frac{T}{2^\ell}, \frac{T}{2^{\ell-1}} \right) \right\}$ to consist of the coordinates $i \in [n]$ with frequency in the interval $\left[ \frac{T}{2^\ell}, \frac{T}{2^{\ell-1}} \right)$. Let $M_\ell = \sum_{i \in \Gamma_\ell} f_i$ be the sum of the contributions of the items in level set $\Gamma_\ell$. Let $\Gamma$ be the set of all coordinates with value greater than 1.

For each coordinate $i \in M_\ell$, $i$ is sampled with probability $p_\ell = \frac{1}{2^{\beta_\ell}}$, where

$$\beta_\ell = \max \left( 0, \ell - \log \left( 10 \left( \frac{\sqrt{C}}{\varepsilon} \right) \log \frac{n}{\varepsilon} \right) \right).$$

Therefore, we consider casework on whether $\ell \leq \log \left( \frac{10}{\varepsilon} \log \frac{n}{\varepsilon} \right)$ or whether $\ell > \log \left( \frac{10}{\varepsilon} \log \frac{n}{\varepsilon} \right)$.

We first consider the first case, where $\ell \leq \log \left( \frac{10}{\varepsilon} \log \frac{n}{\varepsilon} \right)$, so that $p_\ell = 1$. We have $i \in S_\ell$ for any $i \in \Gamma_\ell$. Let $\mathcal{G}_r$ denote the probability that $i$ is not hashed by the $r$-th instance of COUNTSKETCH to a bucket containing any of the other items in $\Gamma$, so that we have $\mathbf{Pr}[\mathcal{E}_r] \geq \frac{2}{3}$ since $|\Gamma| \leq C \leq \frac{1}{\varepsilon}$ and we use $B = \frac{1}{\varepsilon} \cdot \text{polylog} \left( \frac{n}{\varepsilon} \right)$ buckets in each instance of COUNTSKETCH. Then let $\mathcal{E}$ denote the probability that for all items $i \in S_\ell \cap \Gamma_\ell$, there exist $O(\log n)$ instances of COUNTSKETCH such that $i$ is not hashed to a bucket containing any of the other items in $\Gamma_\ell \cap S_\ell$, so that we have $\mathbf{Pr}[\mathcal{E}] = \mathbf{Pr}[\mathcal{G}_1 \vee \mathcal{G}_2 \vee \ldots] \geq 1 - \frac{1}{\text{poly}(n)}$. Conditioning on $\mathcal{G}_r$, the variance for the estimation of $f_i$ by the $r$-th instance COUNTSKETCH is at most $\frac{1}{B} \cdot \frac{100}{\varepsilon^2}$. Since $f_i \geq \frac{T}{2^{\ell-1}} \geq \frac{10}{\varepsilon} \log \frac{n}{\varepsilon}$ then COUNTSKETCH reports $f_i$ as a heavy-hitter with probability $\frac{2}{3}$. Thus, $i$ is reported as a heavy-hitter by some instance of COUNTSKETCH with high probability. It follows by a union bound over all $i \in S_\ell$ that $\widehat{M_\ell} = M_\ell$ high probability.

In the other case, we have $\ell > \log \left( \frac{10}{\varepsilon} \log \frac{n}{\varepsilon} \right)$, so that $p_\ell = \frac{1}{2^{\beta_\ell}}$, where

$$\beta_\ell = \max \left( 0, \ell - \log \left( \frac{10}{\varepsilon} \log \frac{n}{\varepsilon} \right) \right).$$

For any $i \in \Gamma_\ell$, we have $i \in S_\ell$ with probability $p_\ell$. Define $\mathcal{E}_1$ to denote the event that $F_0(S_\ell) \leq (10 \log n) \cdot p_\ell \cdot F_0(S)$ so that $\mathbf{Pr}[\mathcal{E}_1] \geq 1 - \frac{1}{10n^2}$. Let $\mathcal{G}_r$ denote the event that $i \in S_\ell$ is not hashed by the $r$-th instance of COUNTSKETCH to a bucket containing any of the other items in $\Gamma$, so that we have $\mathbf{Pr}[\mathcal{E}_r] \geq \frac{2}{3}$ since $|\Gamma| \leq C \leq \frac{1}{\varepsilon}$ and we use $B = \frac{1}{\varepsilon} \cdot \text{polylog} \left( \frac{n}{\varepsilon} \right)$ buckets in each instance of COUNTSKETCH. Moreover, let $\mathcal{E}_2$ denote the probability that for all items $i \in S_\ell \cap \Gamma_\ell$, there exist $O(\log n)$ instances of COUNTSKETCH such that $i$ is not hashed to a bucket containing any of the other items in $\Gamma_\ell \cap S_\ell$. Then we have $\mathbf{Pr}[\mathcal{E}_2] = \mathbf{Pr}[\mathcal{G}_1 \vee \mathcal{G}_2 \vee \ldots] \geq 1 - \frac{1}{\text{poly}(n)}$. Conditioning on $\mathcal{E}_1$ and $\mathcal{E}_r$, the variance for the estimation of $f_i$ by the $r$-th instance COUNTSKETCH is at most $\frac{1}{B} \cdot \frac{100}{\varepsilon^2} \cdot (10 \log n) \cdot p_\ell$. Since $f_i \geq \frac{T}{2^{\ell-1}} \geq \frac{10}{\varepsilon} \log \frac{n}{\varepsilon}$ then COUNTSKETCH reports $f_i$ as a heavy-hitter with probability at least $\frac{2}{3}$. Hence,

$$\mathbb{E}\left[ \widehat{M_\ell} \right] = \frac{1}{p_\ell} \sum_{j \in \Gamma_\ell} p_\ell \cdot f_j = M_\ell.$$

Moreover,

$$\mathbb{E}\left[ (\widehat{M_\ell})^2 \right] \leq \frac{1}{p_\ell^2} \sum_{j \in \Gamma_\ell} p_\ell \cdot f_j^2 \leq |\Gamma_\ell| \cdot \frac{T^2}{2^{2\ell}} \cdot 2^\ell.$$

Now, we have

$$|\Gamma_\ell| \leq |\Gamma| \leq C \leq \frac{1}{\varepsilon},$$

so that for some large constant $\gamma > 1$,

$$\mathbb{E}\left[(\widehat{M_\ell})^2\right] \leq \frac{1}{\gamma \varepsilon^2 \log^2\left(\frac{n}{\varepsilon}\right)} \leq \frac{\varepsilon^2}{\gamma \log^2\left(\frac{n}{\varepsilon}\right)} \cdot F_0^2(S).$$

Thus by Chebyshev's inequality, we have that with probability at least $1 - \frac{1}{100 \log \frac{n}{\varepsilon}}$,

$$|M_\ell - \widehat{M_\ell}| \leq \frac{\varepsilon}{L} \cdot F_0.$$

The result then follows from union bounding over all $L$ level sets $\Gamma_1, \ldots, \Gamma_L$. $\qquad\square$

We now give the full guarantees of Algorithm 5.

**Theorem D.12.** *Given a stream $S$, an accuracy parameter $\varepsilon \in (0, 1)$, a parameter $C \leq \frac{1}{\varepsilon} \cdot F_0(S)$ for the number of coordinates with frequency more than $1$, and a number of distinct elements that is at least $F_0(S) = \Omega\left(\frac{1}{\varepsilon^2}\right)$, there exists a two-pass streaming algorithm that uses $\frac{1}{\varepsilon} \cdot \operatorname{polylog}\left(\frac{n}{\varepsilon}\right)$ bits of space and provides a $(1 + \varepsilon)$-approximation to the number of distinct elements in the stream with probability at least $0.98$.*

*Proof.* The proof of correctness follows from Lemma D.11. The space complexity follows from the fact that we maintain $B$ buckets in each of the $O(\log n)$ instances of COUNTSKETCH, for $B = \frac{1}{\varepsilon} \cdot \operatorname{polylog}\left(\frac{1}{\varepsilon}\right)$. $\qquad\square$

We now show that any algorithm for $(1 + \varepsilon)$-approximation to distinct elements requires $\Omega\left(\frac{1}{\varepsilon}\right)$ bits of space, even when $C < \frac{1}{\varepsilon}$.

**Theorem D.13.** *For any frequency vector that has the number $C = O\left(\frac{1}{\varepsilon}\right)$ of coordinates with frequency more than $1$, any one-pass streaming algorithm for $(1 + \varepsilon)$-approximation of the number of distinct elements must use $\Omega\left(\frac{1}{\varepsilon}\right)$ bits of space.*

*Proof.* Consider an instance of SetDisj that has $n = \Theta\left(\frac{1}{\varepsilon}\right)$ coordinates. Note that for the purposes of the proof, it suffices to assume that $C = O\left(\frac{1}{\varepsilon^2}\right)$. Namely, let $X$ be the input vector to Alice and let $Y$ be the input vector to Bob. Then $Z := X + Y$ has at most single coordinate with frequency more than $1$. Moreover, any $(1 + \varepsilon)$-approximation to $F_0(Z)$ will distinguish whether the $X$ and $Y$ are disjoint, since we can also compute $F_1(Z)$. Therefore, such an algorithm can be used to solve SetDisj on $\Theta\left(\frac{1}{\varepsilon}\right)$ coordinates and must use space $\Omega\left(\frac{1}{\varepsilon}\right)$. $\qquad\square$

