# OpenReview forum: "On Fine-Grained Distinct Element Estimation"
_ICML.cc/2025/Conference — ICML 2025 poster_

### Official Review · Reviewer_GxnD · 2025-02-15

**Overall Recommendation:** 3

**Summary:**

This paper considers the distinct elements problem in a distributed setting. There are $\alpha$ servers and a universe of $n$ elements, represented as a frequency vector $S$ over all elements in the universe. Each server has a subset $S_i$ of the items. The goal is for the server to send messages to a coordinator such that the number of non-zero entries of $S:=\cup S_i$, denoted by $F_0(S)$ are estimated up to a multiplicative $(1\pm \varepsilon)$ factor.

The authors give a messaging protocol that, assuming an upper bound of $\beta$ on the sum of overlaps $|S_i\cap S_j|$ over all pairs, uses $O(\alpha\log n\log\log n + \sqrt{\beta} \varepsilon^{-2}\log n)$ bits, bypassing the $O(\alpha(\log n + \varepsilon^{-2}))$ worst case lower bound. The authors also offer further improvements if the number if the number of pairwise collisions is promised to be less than $F_0(S)$. In addition, the authors show that in the regime where there results are interesting, that is in the regime where $\beta$ is small compared to $\alpha$, the dependency on $\beta$ achieved by their algorithms is also necessary. Finally, the authors also showcase that their ideas can also be extended to other fields, most notably streaming algorithms where they achieve a multi-pass streaming algorithm with only $\text{polylog}(\varepsilon^{-1},\log n)$ bits of space, as opposed to the polynomial dependencies on $\varepsilon^{-1}$ that are typically necessary.

The algorithms use subsampling ideas to first obtain a coarse estimation of $F_0(S)$ and then likewise to refine the estimation to a the desired $(1\pm \varepsilon)$ factor.

**Claims And Evidence:**

-

**Essential References Not Discussed:**

The related work is reasonable, so missing the following reference did not affect my rating of the paper.

Mridul Nandi, N. V. Vinodchandran, Arijit Ghosh, Kuldeep S. Meel, Soumit Pal, Sourav Chakraborty:
Improved Streaming Algorithm for the Klee's Measure Problem and Generalizations. APPROX/RANDOM 2024

It also uses subsampling as opposed to sketching ideas for distinct elements.

**Experimental Designs Or Analyses:**

-

**Methods And Evaluation Criteria:**

-

**Other Comments Or Suggestions:**

-

**Other Strengths And Weaknesses:**

I think that the algorithm would be a lot more convincing if it recovered the worst case bounds, or were at least competitive with the worst case algorthms without the assumption that $\beta$ is small. Additonally, most beyond worst case results analyze an algorithm that is already widely used. In this case a worst-case performance is not good and the algorithm is designed from scratch.

The modelling of the problem and determining the parameterization in terms of $\beta$ has value. Unfortunately, the ideas themselves and the analysis is not too novel. The techniques have been around for quite a while and while piecing them together is not trivial, I also did not feel like I learned much when reading this paper.

I feel it is a borderline paper, which I rounded up to a weak accept.

**Questions For Authors:**

-

**Relation To Broader Scientific Literature:**

The distinct elements problem is very well studied, but now in the classic sense effectively closed. Some research on this area is still active, but those results typically focus on more general problems.

The authors show that the worst case bound of $\Theta(\alpha (\log n + \varepsilon^{-2}))$ can be improved and that the assumption that they are making is, in some sense, necessary.

**Theoretical Claims:**

The proofs in the main body are correct. I only skimmed the results in the supplementary material, but the proof of the first streaming algorithm is also correct.

---

> ### Author Rebuttal · Authors · 2025-04-01
>
> We thank the reviewer for their feedback and constructive criticism.
>
> > Mridul Nandi, N. V. Vinodchandran, Arijit Ghosh, Kuldeep S. Meel, Soumit Pal, Sourav Chakraborty: Improved Streaming Algorithm for the Klee's Measure Problem and Generalizations. APPROX/RANDOM 2024
>
> > It also uses subsampling as opposed to sketching ideas for distinct elements.
>
> Thank you for pointing out this reference. We have added a reference as an additional work that uses subsampling.
>
> > I think that the algorithm would be a lot more convincing if it recovered the worst case bounds, or were at least competitive with the worst case algorthms without the assumption that $\beta$ is small. Additonally, most beyond worst case results analyze an algorithm that is already widely used. In this case a worst-case performance is not good and the algorithm is designed from scratch.
>
> We emphasize that our bounds are competitive with the worst case bounds. We can further optimize the $O(\alpha\log n\log\log n)$ term to $O(\alpha\log n)$ by referencing an existing technique instead of Algorithm 1, so that overall the only difference between our bounds and the worst-case bounds is the $O(\log n)$ communication required by sending the identity of each sample.
>
> On the other hand, many datasets often exhibit some sort of skewed distribution and our results show that the communication can be significantly improved upon known bounds in these settings.
>
> > The modelling of the problem and determining the parameterization in terms of $\beta$ has value. Unfortunately, the ideas themselves and the analysis is not too novel. The techniques have been around for quite a while and while piecing them together is not trivial, I also did not feel like I learned much when reading this paper.
>
> We acknowledge that the core techniques used (e.g., subsampling, sketching primitives adapted for communication) are built upon existing foundational work in streaming and communication complexity. However, we believe the novelty lies in: 1) Identifying the collision parameter C/β as the key factor governing complexity beyond the worst case. 2) Designing protocols specifically tailored to leverage this parameter. 3) Providing a complete analysis including matching upper and lower bounds in this parameterized setting, which requires non-trivial adaptation and combination of existing techniques. While the tools may be familiar, establishing the tight parameterized complexity is our core technical contribution. We would also like to highlight novel techniques, such as using robust statistics to achieve our streaming algorithm for distinct elements, which allows for accurate estimation even in the presence of adversarial noise and memory constraints; to the best of our knowledge, no prior results utilize robust statistics for distinct element estimation in the streaming setting.
>
> Additionally, although there is a decent body of literature on distributional assumptions and fine-grained complexity, surprisingly there is comparatively much less literature for sublinear algorithms. In particular, there has recently been a number of works in the area of learning-augmented algorithms that specifically use distributional assumptions in addition to machine-learning advice to achieve better guarantees. Thus although distributional assumptions have been studied in the past, we respectfully believe there is value in exploring these directions in new areas, particularly for the purposes of bridging practical algorithms and impossibility results in theory.

---

### Official Review · Reviewer_VuAd · 2025-03-05

**Overall Recommendation:** 4

**Summary:**

The submission provides a more detailed theoretical analysis of the Distributed Distinct Element Estimation problem and shows that under certain assumptions on the distribution of the data points across servers (in particular concerning the number of collisions), previous lower bounds can be overcome. The main contributions include a new theoretical algorithm/protocol with more refined guarantees and an experimental evaluation.

**Claims And Evidence:**

Yes.

**Essential References Not Discussed:**

N/A.

**Experimental Designs Or Analyses:**

I did not spot any issues with the experimental design.

**Methods And Evaluation Criteria:**

Yes.

**Other Comments Or Suggestions:**

N/A.

**Other Strengths And Weaknesses:**

The conceptual idea behind the submission is a good one: there are well-established lower bounds for Distributed Distinct Element Estimation, but these assume certain distributions and it is natural to ask whether the lower bounds can be circumvented when the distributions are closer to those that may occur in some real-world scenarios. The authors show that this is indeed the case. They also provide lower bounds which show that under the considered assumptions, their results are basically tight. The proofs are non-trivial, and overall the contributions are sufficient and in line with what I would expect from a primarily theoretical ICML paper.

The problem itself also seems well-studied, although I am wondering whether it is still relevant in contemporary ML applications. If the authors are aware of more recent applications of the problem in ML, they are welcome to provide some. The paper is also well-written and nicely discusses the implications / context of the obtained results.

## Update after rebuttal
I maintain my assessment and score.

**Questions For Authors:**

In the algorithmic contributions (Theorem 1.1 and 1.2), what would happen if we mis-estimated the number of collisions? In other words, is there any way of using the obtained results when one is not certain about the number of collisions (e.g., can one "test" different choices)?

**Relation To Broader Scientific Literature:**

There is a well-documented connection to previous work.

**Theoretical Claims:**

I did not spot any issues with the theoretical claims.

---

> ### Author Rebuttal · Authors · 2025-04-01
>
> We thank the reviewer for their positive assessment and thoughtful questions.
>
> > The problem itself also seems well-studied, although I am wondering whether it is still relevant in contemporary ML applications. If the authors are aware of more recent applications of the problem in ML, they are welcome to provide some.
>
> Thank you for this question. While Distributed Distinct Elements is a classic problem, variations and related counting/frequency estimation tasks remain highly relevant in modern large-scale ML and data analysis. Examples include: estimating cardinalities in federated learning settings without sharing raw data, analyzing feature overlap or user reach across distributed datasets/services, monitoring network traffic statistics (unique flows/IPs), and estimating distinct items in large graphs or databases distributed across clusters. We will add references or discussion points highlighting these contemporary ML applications in the revised introduction or related work.
>
> > In the algorithmic contributions (Theorem 1.1 and 1.2), what would happen if we mis-estimated the number of collisions? In other words, is there any way of using the obtained results when one is not certain about the number of collisions (e.g., can one "test" different choices)?
>
> Our algorithm is fairly robust, as the error in the estimate for the number of collisions can be translated to an additional number of samples and thus communication. For instance, if the estimate is incorrect by a $O(\log n)$ factor, we can still handle this using an extra $O(\log n)$ factor in the communication. In general, if $\beta$ is underestimated, the protocol's correctness guarantee may fail. If $\beta$ is overestimated, the protocol remains correct but uses more communication than necessary. On the other hand, it's not quite clear how to test different choices, because an incorrect choice could erroneously lead to an estimate that "looks" correct for the incorrect choice.

---

### Official Review · Reviewer_iXZ8 · 2025-03-12

**Overall Recommendation:** 3

**Summary:**

The paper studies distinct element estimation in a distributed setting, where $\alpha$ servers each hold a subset of elements from $[n]$. The goal is to compute the total number of distinct elements approximately while minimizing communication cost.

For a $(1 + \epsilon)$-approximation, prior works establish tight bounds of $\Theta(\alpha / \epsilon^2 + \alpha \log n)$, assuming a constant fraction of elements appear in a constant fraction of servers.

This paper explores a setting where most elements are not widely replicated across servers. Under the assumption that the number of pairwise collisions $C$ satisfies $C = \beta \cdot O(\min (F_0(S), 1 / \epsilon^2 ))$, where $F_0(S)$ denotes the number of distinct elements, the authors improve the upper bound to $O(\alpha \log n \log \log n + \sqrt{\beta} (\log n) \cdot \min (F_0(S), 1/ \epsilon^2))$, with further improvements when $C < F_0(S)$. The paper also establishes matching lower bounds.

Algorithms 1 and 2 build on standard techniques, but their analysis leverages the newly introduced pairwise collision parameter.

**Claims And Evidence:**

Yes, the claims are supported by mathematical guarantees.

**Essential References Not Discussed:**

N/A

**Experimental Designs Or Analyses:**

Experiments contain validation of the theoretical setting.

**Methods And Evaluation Criteria:**

Yes

**Other Comments Or Suggestions:**

1. See Theoretical Claims.
2. Line 124, $C = O(\alpha)$? Maybe something is missing there.

**Other Strengths And Weaknesses:**

The paper utilizes the input pattern to improve the current bounds.

**Questions For Authors:**

See Theoretical Claims.

**Relation To Broader Scientific Literature:**

The paper falls into the category of exploiting data patterns to overcome existing theoretical bounds.

**Theoretical Claims:**

Is Algorithm 1 necessary? As stated in Line 58 on Page 2, a one-pass streaming algorithm for distinct element estimation can be transformed into the distributed setting, yielding a protocol with $O(\alpha / \epsilon^2 + \alpha \log n)$ bits of communication.

Since Algorithm 1 aims for a constant-factor approximation of the number of distinct elements, one could instead apply the aforementioned protocol, achieving a communication cost of $O(\alpha + \alpha \log n)$, which appears lower than that of Algorithm 1.

---

> ### Author Rebuttal · Authors · 2025-04-01
>
> We thank the reviewer for their positive feedback and the pertinent question regarding Algorithm 1.
>
> > Is Algorithm 1 necessary? As stated in Line 58 on Page 2, a one-pass streaming algorithm for distinct element estimation can be transformed into the distributed setting, yielding a protocol with $O(\alpha/\epsilon^2 +\alpha\log n)$ bits of communication.
>
> > Since Algorithm 1 aims for a constant-factor approximation of the number of distinct elements, one could instead apply the aforementioned protocol, achieving a communication cost of $O(\alpha +\alpha\log n)$, which appears lower than that of Algorithm 1.
>
> This is a great point. We originally had Algorithm 1 as a warm-up to the main results in Algorithm 2 and Algorithm 3 because both the algorithmic structure and the corresponding analysis are similar but simpler. However, given the additional $\log\log n$ factor incurred by Algorithm 1, we agree that it would be better to simply reference an existing protocol that achieves communication cost $O(\alpha+\alpha\log n)$. Thanks for your suggestion!
>
> > Line 124, $C=O(\alpha)$? Maybe something is missing there.
>
> We have corrected the typo to be $C=O(\alpha)\cdot F_0(S)$, consistent with the implications of the statement -- thanks for pointing this out.

---

### Official Review · Reviewer_XJEv · 2025-03-13

**Overall Recommendation:** 3

**Summary:**

In this paper the problem of distinct element estimation is studied. More precisely, we are given an universe [n] and $\alpha$ servers and each of the servers receives a subset of the universe. Now, the goal is to compute a $(1+\epsilon)$-approximation of the number of distinct elements using minimal communication among the servers.
This problem is well-studied and worst case bounds exists.
These bounds, however, are  based on assumptions which do not hold in practice.
Thus, this problem is studied parameterized by the number of pairwise collisions.
Using this parameter the paper presents a protocol with a low number of bits for communication, breaking previous lower bounds if the parameter is small.
Also, matching lower bound under this parameter a presented.
Finally, a proof-of-concept implementation is provided which shows the effectiveness of the new protocol.

**Claims And Evidence:**

I think all claims in the paper are convincing.

**Essential References Not Discussed:**

No, see Relation To Broader Scientific Literature

**Experimental Designs Or Analyses:**

Yes, I checked the experimental details. I am convinced by the result.
I only think that more data should be used to obtain higher quality results (see Methods And Evaluation Criteria). However, I don't think that this would change the results drastically.

**Methods And Evaluation Criteria:**

For the theory, I say yes.

For the experiments, I think the data set is quite small. I think without too much effort more data sets can be generated, for example by considering each batch of 1 million events individually. This then results in 40 times as many data sets.
I don't think that this is a big issue since the main focus of the paper is theory and the experiments so far already show that the new approach is very good in practice.

**Other Comments Or Suggestions:**

l 87 c 1: please define zipifan distribution

l 124-129 c1: please provide an explanation for this behavior

l 237 c2: constant factor >> 4-approx

l 670: what is P,Q?

l 678: I think this formal description of the problem should be part of the main body

l 750: Here your arguments were to short for me to understand the proof as a non expert. Providing more details here would be very helpful for me. For example instead of ``using standard expectation and variance techniques'' please provide the precise arguments.
(similar in l 770)

l 797L where does the factor 100 come from?

l 866: this seems natural; but please provide the padding argument here

l 877: why ``Unfortunately''? I think this is a positive result

l 974: please provide a reference for this lower bound

appendix C.1 had some nice motivation; I think this should also be mentioned in the main body

**Other Strengths And Weaknesses:**

combining parameterized algorithmics and distinct element estimation is a natural idea and shows the future potential of this approach for other related problems

**Questions For Authors:**

Q1: You say that the new protocol is better if C is small. But this argument only takes the second summand into account. In your new bound the first summand is larger than the first summand in the old bound. Why is this first summand not important/dominated by the second?

Q2: I don't understand why for non-binary vectors you want $v_i^{(a)}\ge 1$ and $v_i^{(b)}\ge 1$. For me, it seems more natural to require $v_i^{(a)}= v_i^{(b)}$. Does this make the problem significantly more complicated? Is this problem relevant in practice? Is this studied?

Q3: In most theorems you use completely different probabilities. Why is this the case? Is there an easy argument that any constant probability can be used? If yes, please provide such an argument.

**Relation To Broader Scientific Literature:**

The key idea is to use the parameter C of pairwise collisions. This is a new idea and this parameter is small in practice.
For the unparameterized setting the authors cite all relevant literature as far as I can tell.

**Theoretical Claims:**

Since I am not an expert in the field, I only checked some small proofs and tried to understand the ideas of the proof. I skipped most proofs in the appendix.

---

> ### Author Rebuttal · Authors · 2025-04-01
>
> We thank the reviewer for their detailed feedback and insightful questions.
>
> > For the experiments...more data sets can be generated
>
> We agree that increasing the number of datasets could provide additional empirical validation. Given the structure of the data, we anticipate that the overall pattern of results would remain similar. However, we see the value in this approach and will explore generating additional datasets to further strengthen our experimental evaluation
>
> > l 87 c 1: please define zipifan distribution
>
> We have added the formal definition of Zipfian distribution in the preliminiaries and provided a forward pointer at this location.
>
> > l 124-129 c1: please provide an explanation for this behavior
>
> We have corrected the typo to be $C=O(\alpha)\cdot F_0(S)$, consistent with the implications of the statement -- thanks for pointing this out!
>
> > l 237 c2: constant factor >> 4-approx
>
> We have specified this in the algorithm.
>
> > l 670: what is P,Q?
>
> $P$ and $Q$ are the two distributions $\mu_1$ and $\mu_2$ -- we have unified the notation now.
>
> > l 678: I think this formal description of the problem should be part of the main body
>
> Thanks for the suggestion, we have moved this formal description to the main body in Section 2 and swapped out some of the full proofs to the appendix.
>
> > l 750: Here your arguments were to short for me to understand the proof as a non expert...please provide the precise arguments. (similar in l 770)
>
> We have adjusted to language to clarify that the estimator provides an unbiased estimate to the total number of distinct elements and because of the number of samples, the variance is also sufficiently small. Then by applying a concentration inequality such as Chebyshev's inequality, it follows that the resulting estimator provides a $(1+\epsilon)$-approximation to the total number of distinct elements.
>
> > l 797L where does the factor 100 come from?
>
> The number 100 comes from being a sufficiently large constant to apply the concentration inequalities that would result in a $(1+\epsilon)$-approximation. We have clarified this.
>
> > l 866: this seems natural; but please provide the padding argument here
>
> We have added a description of the padding argument to the corresponding location in the appendix.
>
> > l 877: why ``Unfortunately''? I think this is a positive result
>
> This is a positive result, but it rules out the proposed construction for a  "hard distribution" toward the desired lower bounds, which is arguably unfortunate in the given context. We have rephrased this language.
>
> > l 974: please provide a reference for this lower bound
>
> Thanks, we have added the reference to this lower bound, appearing in Jayram and Woodruff (2013).
>
> > appendix C.1 had some nice motivation; I think this should also be mentioned in the main body
>
> Although there is insufficient room in the main body to include the entire motivation, we have added some of the motivation to the main body and included a pointer to Appendix C.1 for additional discussion.
>
> > Q1: You say that the new protocol is better if C is small. But this argument only takes the second summand into account. In your new bound the first summand is larger than the first summand in the old bound. Why is this first summand not important/dominated by the second?
>
> Our first summand is $O(\alpha\log n\log\log n)$ while the first summand in the old bound is $O(\alpha\log n)$. Due to the small $O(\log\log n)$ factor, we did not optimize the first summand.
>
> In fact, this discrepancy is due to Algorithm 1, which we included because it leads to a natural description of Algorithm 2 and 3. Instead, we can also reference an existing procedure so that our first summand becomes $O(\alpha\log n)$, without changing any other terms in our bound. Thus, we match the first summand of the old bound over all regimes. This was also picked up by Reviewer iXZ8, who pointed out the connection.
>
> > Q2: I don't understand why for non-binary vectors you want $v_i^{(a)}\ge 1$ and $v_i^{(b)}\ge 1$. For me, it seems more natural to require $v_i^{(a)}=v_i^{(b)}$. Does this make the problem significantly more complicated? Is this problem relevant in practice? Is this studied?
>
> Intuitively, an item $i$ is shared across multiple servers $a$ and $b$ if $v_i^{(a)}\ge 1$ and $v_i^{(b)}\ge 1$. For the purposes of the distinct elements problem, there is no difference from the perspective of a server $a$ if $v_i^{(a)}=1$ or $v_i^{(a)}>1$ because either way, the server $a$ knows that $i$ has a non-zero count.
>
> > Q3: In most theorems you use completely different probabilities. Why is this the case? Is there an easy argument that any constant probability can be used? If yes, please provide such an argument.
>
> Yes, any constant probability larger than $\frac{1}{2}$ can generally be used. This is because by taking $O\left(\log\frac{1}{\delta}\right)$ independent instances of an algorithm and computing the median, the probability of success is boosted to $1-\delta$.

---

> > ### Comment · Reviewer_XJEv · 2025-04-03
> >
> > Thanks for your answers!
> >
> > About Q3: Thanks, this is what I expected. In my opinion it is slightly better to use the same probability in each statement to avoid confusion.
> >
> > About the experiments: My intuition is the same that more data generated from this data set should yield the same result. But nonetheless it is better to have such an evaluation.
> >
> > Hence, I will keep my current score.

---

> > > ### Author Response · Authors · 2025-04-05
> > >
> > > We thank the reviewer for the additional feedback and the continued correspondence.
> > >
> > > > About Q3: Thanks, this is what I expected. In my opinion it is slightly better to use the same probability in each statement to avoid confusion.
> > >
> > > We have unified each statement to have the same probability $\frac{2}{3}$ across the document.
> > >
> > > > About the experiments: My intuition is the same that more data generated from this data set should yield the same result. But nonetheless it is better to have such an evaluation.
> > >
> > > Following the reviewer suggestion, we have generated $20$ datasets from this larger dataset and conducted experiments on these datasets. The behavior is mostly the same, so that when each server is expected to send only 16 samples (or more), the worst performance across the 20 iterations is roughly 92% accuracy, while the average is closer to 95%.
> > >
> > > The line plots for the worst, average, and best case performances across the 20 iterations are available at https://anonymous.4open.science/r/parameterized-distributed-distinct-elements-8E7D/samples-vs-err-central.png
> > >
> > > We have also uploaded the entire code to the anonymous repository at https://anonymous.4open.science/r/parameterized-distributed-distinct-elements-8E7D/ (though the csv file from CAIDA itself is too large to upload)

---

### Decision · Program_Chairs · 2025-05-01

**Decision:**

Accept (poster)

**Comment:**

Reviewers generally appreciate the conceptual message of this paper, particularly the idea of leveraging dataset properties in practice to bypass worst-case lower bounds. While some noted that the algorithmic ideas and analysis lack significant novelty, they found them generally acceptable. Please consider to incorporate the following reviewer suggestions into the next version of the paper: (1) Remove Algorithm 1 to streamline the presentation. (2) (Optional) develop a method for learning the number of pairwise collisions to ensure that the worst-case bounds asymptotically align with the state-of-the-art across all parameter ranges.